# Green water availability and water-limited crop yields under a changing climate in Ethiopia

Mosisa Tujuba Wakjira[1], Nadav Peleg[2], Johan Six[3], Peter Molnar[1]

[1]Institute of Environmental Engineering, ETH Zurich, Laura-Hezner-Weg 7, CH-8093 Zürich, Switzerland

[2]Institute of Earth Surface Dynamics, University of Lausanne, CH-1015 Lausanne, Switzerland

[3]Institute of Agricultural Science, ETH Zurich, Universitätstrasse 2, CH-8092 Zürich, Switzerland

*Correspondence to*: Mosisa Tujuba Wakjira (mosisatujuba@gmail.com; wakjira@ifu.baug.ethz.ch)

**Abstract.** Climate change is expected to influence future agricultural water availability, posing particular challenges in rainfed agricultural systems. This study aims to analyze the climatology of green water availability and water-limited attainable yields
(AY) – the maximum crop yield achieved with available green water under optimal soil nutrient and crop management, considering four major cereal crops (teff, maize, sorghum, and wheat) produced in Ethiopia. An agrohydrological modelling framework was developed to simulate climatic-hydrological-crop interactions. The model was applied to a reference (1981-2010) and future (2020-2099) periods under low, intermediate, and high greenhouse gas emission scenarios to: (i) evaluate the current green water availability and AY potential; (ii) assess their climate-driven changes; and (iii) analyze the sensitivity of
changes in AY to changes in rainfall and atmospheric evaporative demand. With regional variations based on climatic regimes, the main growing season (Meher, May to September) has an average AY of 79 % of a fully irrigated potential yield with an average soil moisture deficit of 29 % of moisture content at full water holding capacity. AY of the short growing season (Belg, February-May) is on average 37 % of the potential yield, with a soil moisture deficit of 56 %. Under the future climate, Meher is expected to experience small changes in the range of ±5 % with dominantly positive trends in the 2030s and decreases in
the 2060s and 2080s, mainly driven by changes in the atmospheric evaporative demand due to rising temperatures. The Belg-producing regions are expected to experience increased AY that is dominantly controlled by increases in rainfall. On the other hand, a substantial yield gap is identified between actual and water-limited yields. This points to the need for combining green water management practices with nutrient and tillage management, plant protection, and cultivar improvement to close the yield gaps and build up the climate resilience of farmers.

## 1. Introduction

Green water, the infiltrated part of rainfall that is stored in the soil root zone and returns to the atmosphere in the form of evapotranspiration (Falkenmark, 2006), is the sole source of moisture in rainfed agriculture (RFA) systems (Rockström, 1999). Green water accounts for an estimated 80 % of the global agricultural evapotranspiration fluxes, and RFA systems produce ~60 % of the global food (Mekonnen and Hoekstra, 2011; Molden et al., 2011; Rockström et al., 2010; Sposito, 2013). The
reliance on green water varies across regions, with sub-Saharan Africa (SSA) being particularly dependent on this water

resource, where close to 95 % of croplands are under rainfed agriculture practices (Abrams, 2018; Laderach et al., 2021). Due to the highly dynamic nature of green water availability (GWA), influenced by climatic and biophysical factors that vary in space and time, RFA systems are strongly climate-sensitive under moisture-limited conditions (Kang et al., 2009; Meng et al., 2023; Park et al., 2022). Climate change presents an additional major challenge to the system, undermining crop production

with potential consequences of food insecurity, livelihood losses, and economic crises (FAO, 2022).

Previous global-scale assessments have highlighted the challenges posed by climate change on agriculture, both in the past and the future, leading to regionally varying intensification of agricultural water scarcity and a decrease in crop yields (e.g., Borgomeo et al., 2020; Jägermeyr et al., 2021; Lobell et al., 2011; Rosenzweig et al., 2014). This is particularly evident across the tropics including the highly climate-vulnerable SSA countries (Burke et al., 2009; Kummu et al., 2021; Müller et al., 2011;

Rezaei et al., 2023; Rosenzweig et al., 2014; Schlenker and Lobell, 2010). Adaptation actions tailored to specific contexts are now a top priority for mitigating the impacts of climate change-induced water scarcity in rainfed agriculture (RFA) systems, primarily at the national scale through the implementation of National Adaptation Plans (NAP) under the auspices of the United Nations Framework Convention on Climate Change (UNFCCC, 2021) and other programs.

Ethiopia is one of the countries committed to formulating and implementing the NAP, including in agriculture, the sector

which the country and its population heavily rely on (FDRE, 2019). In fact, climate adaptation is a much-needed action in Ethiopian smallholder agriculture, of which ~95 % is dependent on GWA, and at the same time it is an indispensable activity supporting the food and income of nearly 80 % of the population. The NAP and its implementations are, in principle, based on analyses of climate change impacts and vulnerability, along with relevant adaptation actions (UNFCCC, 2021; Warren et al., 2018). Previous studies on the crop yield impacts of climate change in Ethiopia have mainly focused on local case studies

(e.g., Abera et al., 2018; Araya et al., 2015; Degife et al., 2021; Hadgu et al., 2015; Kassie et al., 2014; Markos et al., 2023; Moges and Gangadhara, 2021), making them patchy and insufficient as a comprehensive guide for the NAP. Here we propose a framework that allows for a regional comprehensive analysis of agrohydrological responses to climate change across the RFA region of Ethiopia.

This framework builds on the key factors that influence GWA and their cascading effects on crop yield, to guide appropriate

agricultural water management planning, decisions, and actions. Several factors influence GWA. While rainfall amount and distribution fundamentally determine the green water supply, rainfall event characteristics such as intensity, frequency, and duration, combined with surface biophysical conditions (mainly soil, land cover, terrain slope, and roughness), govern the partitioning of rainfall into overland flow and infiltration (Rockström and Gordon, 2001; Schuol et al., 2008). Soil properties (e.g., textural composition, organic matter content, thickness, and salinity) determine the rainfall partitioning to infiltration and

runoff, the subsurface flow, green water storage capacity, and plant accessibility, while potential evapotranspiration driven by air temperature, radiation, wind, relative humidity, and vegetation, controls the return flow of green water to the atmosphere (Ringersma et al., 2003).

In moisture-limited regions, crop yield is proportional to the magnitude of the evapotranspiration flux, which is constrained by moisture availability (Hatfield and Dold, 2019; Steduto et al., 2012). The maximum yield that can be achieved with the available green water under the best nutrient input and crop management conditions in RFA systems is considered to be the *water-limited attainable yield* (van Ittersum et al., 2013; Lobell et al., 2009). In conditions where water is not a limiting factor, like in RFA systems in humid agroecologies and fully irrigated systems in dry agroecologies, the maximum achievable crop yield is limited by energy, and thus is called *energy-limited yield potential*.

Quantification of the complex interactions between climate, soil, cultivar and crop management is commonly done with mechanistic crop models, for example, APSIM (Holzworth et al., 2014), AquaCrop (Raes et al., 2009; Steduto et al., 2009), CROPSYST (Stöckle et al., 2003), DSSAT (Jones et al., 2003), EPIC (Williams, 1990), and WOFOST (van Diepen et al., 1989), which are used to simulate crop growth and yields at field scale. These models combine hydrological models that simulate hydroclimatic processes to estimate water availability in the root zone, and crop growth models that simulate the crop physiological responses to the agro-environmental and management conditions (Foster and Brozović, 2018; Siad et al., 2019). Model complexity, and associated issues such as high input data and parameter requirements, and computational demand, are major constraints of mechanistic crop models especially when applied in a distributed setting to a large geographic domain, which necessitates a compromise between the level of complexity and the purpose of the modelling (Adam et al., 2011; Ramirez-Villegas et al., 2017).

Here, we employ a simplified agrohydrological modelling framework to assess the impacts of climate change on GWA and the subsequent effects on major cereal crops (teff, maize, sorghum, and wheat) produced in the entire RFA region of Ethiopia. We address three research questions that are relevant for future agricultural water management planning and decision-making in Ethiopia: (i) what is the current green water availability and how will it change under the future climate across the RFA region?; (ii) how will these changes affect water-limited attainable crop yields (AY)?; and (iii) which hydroclimatic factor (rainfall or evapotranspiration) dominantly drives these changes and where? We base our assessment on crop response conditions where all crop growth factors except water are unlimited so that we can capture the effects of climate-driven changes in crop yields in a relative manner. Furthermore, we critically discuss the implications of the changes for water management in the NAP and similar agricultural development initiatives. We also present the workflow of our modelling framework to ensure its robust application in addressing similar or related agrohydrological research questions.

## 2. Methods

### 2.1. Study area

The study area covers the entire rainfed arable parts of Ethiopia (hereafter RFA region), as mapped in previous land use analysis (Kassawmar et al., 2018), encompassing an area of about 667,000 km$^2$ (59 % of the landmass of Ethiopia, see the blue outline in Figure 1a). The climate of the RFA region ranges from humid to semi-arid (Figure 1c) with mean annual potential evapotranspiration ranging from 700 mm in the highland regions to 1800 mm in the western lowland part of the RFA region.

The mean annual rainfall of the RFA region ranges from 270 mm in the eastern part to 2100 mm in the western and southwestern parts (Figure 1b). The rainfall has unimodal patterns in the central, western, northern, and northeastern parts with the highest rainfall in July/August depending on the location (Segele and Lamb, 2005; Wakjira et al., 2021). This rainy season typically spans from May to September, coinciding with the main growing season, locally known as 'Meher', during which approximately 88 % of the total annual crop harvest occurs (CSA, 2007). The southern and eastern parts of the RFA region on

the other hand, largely experience a bimodal pattern with two rainy seasons in spring and autumn. The spring (February-May) rainy period is considered the second and shorter growing season, locally known as 'Belg', in the southern and southeastern part of the RFA region. Cereals account for about 80 % of the crops produced, with teff, maize, sorghum, wheat, and barley being the top varieties, also serving as the main staple food crops in the country (CSA, 2007, 2010).

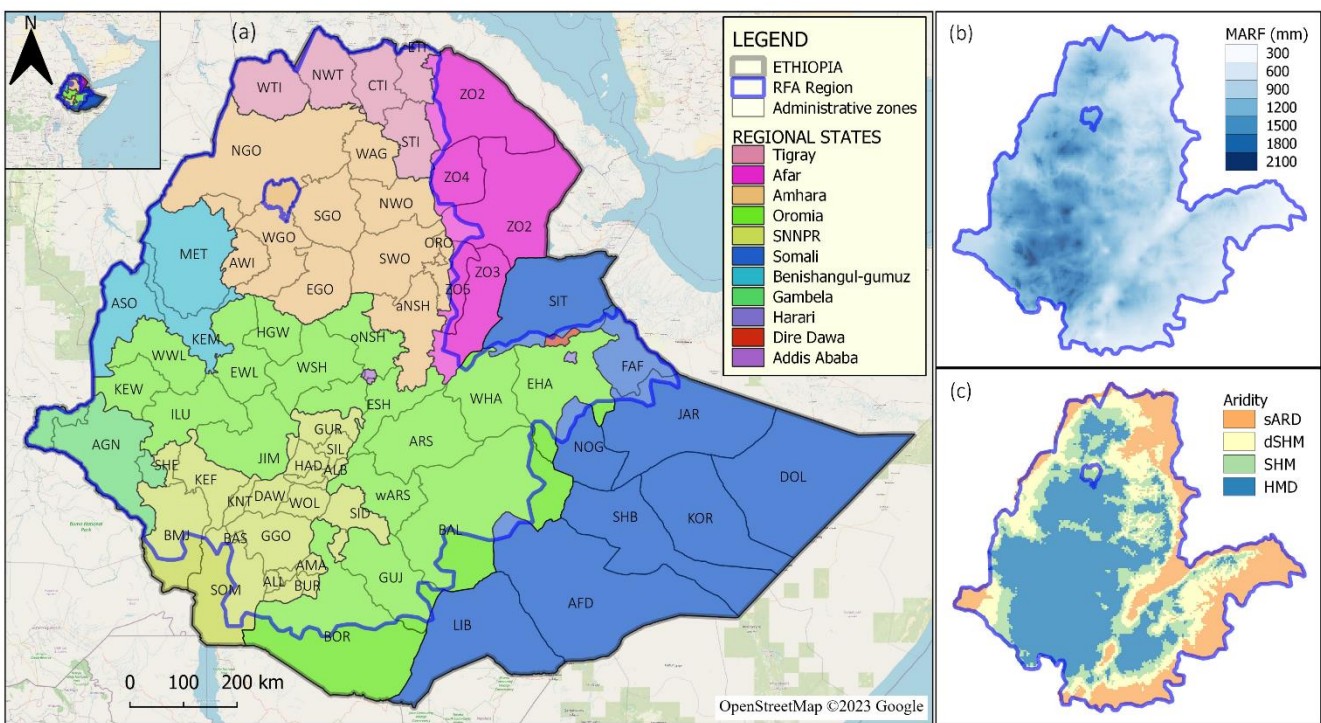

**Figure 1: (a) Map of Ethiopia, the spatial extent of the rainfed agricultural region (blue outline, Kassawmar et al., 2018), administrative zones (designated by short names) in the nine regional states. The complete list of the zones with full names is given in Table S1 of the supplementary materials. (b) Mean annual rainfall (MARF) of the RFA region for the period 1981-2010, based on CHIRPS. (c) Climatic regimes (aridity) of the RFA region (classification based on Spinoni et al., 2015). sARD: semi-arid, dSHM: dry sub-humid, SHM: sub-humid, HMD: humid**

**2.2. Data**

Daily rainfall data for the reference period (1981-2010) was obtained from the Climate Hazards Infrared Precipitation with Station (CHIRPS) dataset (Funk et al., 2015), which is available at 0.05° x 0.05° spatial resolution. Maximum and minimum

daily 2-m air temperature data consists of the bias-corrected and downscaled ERA5-Land (BCE5) dataset (Wakjira et al., 2022, 2023), which is also available at 0.05° x 0.05° spatial resolution. Other climate variables (solar radiation, wind speed, and dew

point temperature) were retrieved from ERA5-Land (Muñoz-Sabater et al., 2021) and disaggregated to a resolution of 0.05° x 0.05° using a bilinear interpolation method.

Future climate data were derived by downscaling multiple Global Circulation Model (GCM) projections, i.e., 26 for precipitation and solar radiation and 21 for maximum and minimum daily temperatures (listed in Table S2), from their native coarse resolution to 0.05° x 0.05° grid resolution using change factor (delta) approach (e.g., Anandhi et al., 2011; Teutschbein

and Seibert, 2012). The future climate scenarios include three Shared Socioeconomic Pathways SSPs (Meinshausen et al., 2020; O'Neill et al., 2016), namely the SSP1-2.6 (low greenhouse gas emission), SSP2-4.5 (intermediate emission) and SSP5-8.5 (high emission).

Gridded soil texture and organic carbon content data were retrieved from SoilGrids (Poggio et al., 2021). Although the SoilGrids soil data is available at 250 m grid resolution, we upscaled the data to 0.05° (~5 km) to harmonize the spatial

resolution with that of the climate datasets. Curve number (CN) values for agricultural land use were obtained from the USDA (1985) lookup table according to the hydrologic soil group dataset developed by Ross et al. (2018).

Independent soil moisture (SM), actual evapotranspiration (ETa), surface runoff, and crop data were utilized to evaluate the agrohydrological model simulations. Four satellite and model-based global SM datasets, ESA CCI (Dorigo et al., 2017), GLDAS 2.1 (Rui and Beaudoing, 2020), LPMR-TMI (Teng and Parinussa, 2021), and SGD-SM (Zhang et al., 2022), were

considered in the evaluation. ETa estimates were retrieved from five products – SSEBop (Senay et al., 2020), PML-v2 (Zhang et al., 2019), MOD16A2 (Mu et al., 2019), GLDAS 2.1 (Rui and Beaudoing, 2020), and TerraClimate (Abatzoglou et al., 2017), covering the period 2003-2010. Surface runoff data were collected from published runoff plot measurements at 20 locations (Table S3) across the RFA region of Ethiopia. Regarding the crop data, paired water-limited (Yw) and potential yields (Yp) data were collected from published case studies that are based on field trials and calibrated crop model simulation

experiments at 26 locations, and from the Global Yield Gap Atlas, GYGA (Global Yield Gap Atlas, 2024), at 19 additional locations across the study area (refer Table S4 for more details). In addition to the yield data, total cereal production (TCP), consisting of the sum of all cereals (maize, teff, sorghum, wheat, barley, millet, oat, and rice) produced in Ethiopia, which was derived from the annual Agricultural Sample Survey (AgSS) reports (e.g., CSA, 2010) gridded by Wakjira et al. (2021) for the period 1995-2010, was also utilized in the evaluation.

**2.3. Agrohydrological modelling**

The agrohydrological modelling framework (Figure 2) that interlinks climate-hydrological-crop (CHC) interactions was developed to simulate the effects of climate change on green water fluxes, and its cascading influences on crop yield. The CHC model consists of three modules for climate, hydrology, and crops.

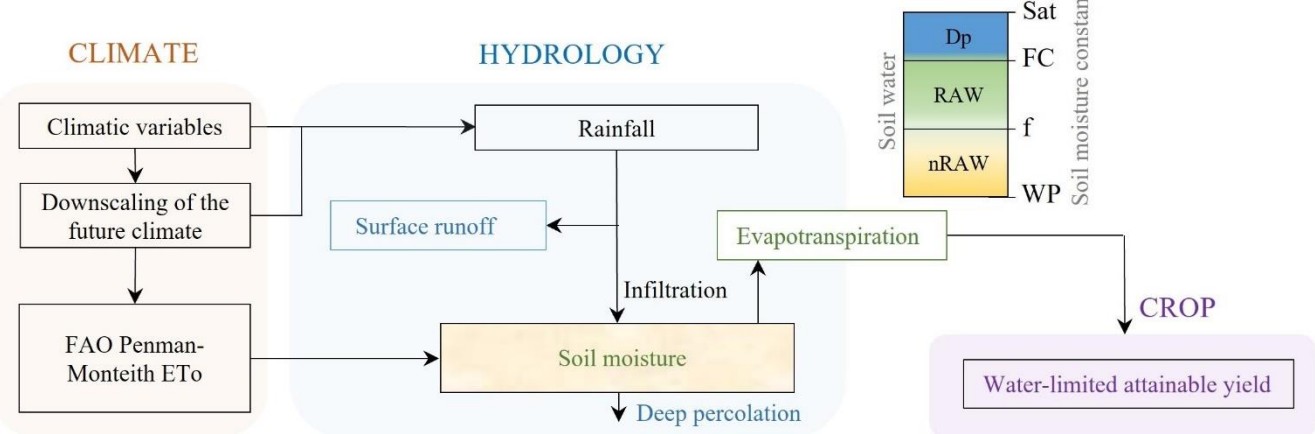

**Figure 2: The CHC model used for the assessment of climate-driven changes in green water availability and water-limited attainable yields. Dp = deep percolation, RAW = readily available water, nRAW = non-readily available water, FC = field capacity, WP = wilting point, f = critical moisture depletion factor.**

### 2.3.1. Climate module

The climate module uses climatic variables to compute the daily reference evapotranspiration $ETo$ [mm d$^{-1}$] using the FAO

Penman-Monteith equation considering a hypothetical reference grass (Allen et al., 1998):

$$ETo = \frac{0.408\Delta(R_n - G) + \gamma\frac{900}{T+273}u_2(e_s - e_a)}{\Delta + \gamma(1 + 0.34u_2)},$$ (1)

where $R_n$ [MJ m$^{-2}$] is net radiation at the crop surface, G [MJ m$^{-2}$] is soil heat flux, T [°C] is 2-m mean daily air temperature, $u_2$ [m s$^{-1}$] is 2-m daily wind speed, $e_s - e_a$ [kPa] is the saturation vapor pressure deficit, $\Delta$ [kPa °C$^{-1}$] is the slope of the vapor pressure curve, and $\gamma$ [kPa °C$^{-1}$] is the psychometric constant.

### 2.3.2. Hydrology module

The hydrology module simulates the dynamic soil water balance without lateral surface and subsurface routing. A conceptual (bucket) water balance model is applied to the soil hydrological fluxes – surface runoff $Q$ [mm], actual evapotranspiration $ETa$ [mm], deep percolation $Dp$ [mm], and changes in soil moisture $\Delta SM$ [mm] from rainfall input $P$ [mm] at daily time step $t$:

$$\Delta SM_t = P_t - Q_t - ETa_t - Dp_t.$$ (2)

Surface runoff is simulated according to the US Soil Conservation Service (SCS) curve number method (USDA, 1985).

$$Q_t = \begin{cases} \frac{(P_t - I_{a,t})^2}{(P_t - I_{a,t}) + S_{20,t}}, & \text{if } P_t > I_a \\ 0 & \text{, otherwise} \end{cases}. \tag{3}$$

where $I_a$ [mm] is an initial abstraction, and $S$ [mm] is the maximum surface retention after runoff begins. We chose the SCS curve number method because it provides a reliable estimate of surface runoff with reduced number of model parameters. This method accounts for soil hydrologic properties, land use, and antecedent soil moisture conditions (AMC) to partition rainfall into infiltration and surface runoff. Consequently, the soil water balance components of widely applied crop models such as DSSAT (Jones et al., 2003), AquaCrop (Hsiao et al., 2009; Raes et al., 2009; Steduto et al., 2009), APSIM (Holzworth et al., 2014), CROPSYST (Stöckle et al., 2003), EPIC (Williams, 1990), and SWAT+ (Bieger et al., 2017; Čerkasova et al., 2023), among others are either entirely built on the SCS curve number method, or provide it as a simpler alternative to infiltration-based models.

In the original SCS curve number model, the initial abstraction ratio of 20 % was assumed so that $I_a$ is 20 % of $S$ (indicated by the subscript in Eq. (3)). However, recent insights have led to an update in the abstraction ratio to 5 %, which yielded higher runoff prediction performance of the SCS model. A proposed conversion function from $S_{20}$ to $S_{05}$ is expressed by Hawkins et al. (2020):

$$S_{05} = 1.42S_{20}. \tag{4}$$

$S$ is determined by the land surface conditions including soil characteristics as represented by the hydrologic soil group, land use land cover (in our case agricultural land), and antecedent soil moisture, all of which are parameterized as a dimensionless curve number ($CN$) value. We estimated $S$ using Eq. (5).

$$S_{20} = 254 \left( \frac{100}{CN} - 1 \right). \tag{5}$$

$CN$ is updated daily for antecedent moisture conditions (AMC), which are often categorized as *dry*, *normal*, and *wet*. The $CN$ values obtained from the USDA lookup table correspond to the *normal* AMC condition. Accordingly, we estimated the CN values for the *dry* and *wet* AMC following Raes et al. (2022) and Smedema and Rycroft (1983):

$$CN_{dry} = 16.91 + 1.348CN - 0.01379(CN)^2 + 0.0001172(CN)^3,$$

$$CN_{wet} = 2.5838 + 1.944CN - 0.014216(CN)^2 + 0.000045829(CN)^3. \tag{6}$$

We then adjusted $CN$ at every time step by linearly interpolating between $CN$ and $CN_{dry}$ or $CN_{wet}$ depending on the soil moisture condition, assuming the soil moisture content corresponds to $\theta_{WP}$ for $CN_{dry}$, $\theta_{0.5(\theta_{FC} + \theta_{WP})}$ for $CN$, and $\theta_{FC}$ for

$CN_{wet}$. Note that $\theta_{FC}$ [m$^3$ m$^{-3}$] and $\theta_{WP}$ [m$^3$ m$^{-3}$] are volumetric moisture contents at field capacity and wilting point and they

were estimated using the widely used pedotransfer function developed by Saxton and Rawls (2006).

Applying the conversion from Eq. (4) for $S_{05}$ and then expressing $I_a$ in terms of $S_{20}$ in Eq. (3), the expression for $Q$ can be rewritten as:

$$Q_t = \begin{cases} \dfrac{(P_t - 0.071S_{20,t})^2}{(P_t + 1.349S_{20,t})} & \text{, if } P_t > 0.071S_{20} \\ 0 & \text{otherwise} \end{cases} \tag{7}$$

The rainfall that is in excess of $Q$ (Eq. (7)) infiltrates into the soil and refills the available soil storage $V_t$[mm] $= 1000Z(\theta_{sat} - \theta_{t-1})$, where $\theta_{sat}$ [m$^3$ m$^{-3}$] and $\theta_{t-1}$ [m$^3$ m$^{-3}$] are volumetric soil moisture content at saturation and on the

previous day (respectively), and $Z$ [m] is the soil depth. We considered that the top 60 cm homogeneous soil layer contains the majority of the root biomass (e.g., Fan et al., 2016; Mthandi et al., 2013), and hence this is the agrohydrological active soil depth Z for the major crops. ETa is determined by moisture availability (green water) relative to the readily available water, $RAW$ [mm] $= 1000Zf(\theta_{FC} - \theta_{WP})$, where $f$ [-] is a critical depletion fraction that represents the soil moisture level below which plants experience moisture stress. For all four cereal crops studied, the value of $f$ was set to 0.55 following Allen et al.

(1998). ETa was computed as $ETa_t = K_{s,t}ET_{o,t}$, where $Ks$ [-] is the soil moisture stress coefficient (Allen et al., 1998):

$$K_{s,t} = \begin{cases} \dfrac{SM_t}{RAW} & \text{if } SM_t < RAW \\ 1 & \text{if } SM_t > RAW \end{cases} \tag{8}$$

If the soil is saturated, the infiltrated water which exceeds the soil field capacity percolates into deeper soil layers at the rate $Dp_t = 1000Z(\theta_t - \theta_{FC})$.

### 2.3.3. Crop module

The crop module simulates water-limited attainable yield (AY) based on the FAO water production function (Doorenbos and

Kassam, 1979). The FAO water production function is a well-established relationship between climate and crop yield, explaining relative yield as a function of the evapotranspiration ratio, i.e., the ratio of actual to potential evapotranspiration on a seasonal scale. It is widely applied to assess the effects of water scarcity and climate change impacts on crop yields to inform agricultural water management planning and decision-making (e.g., Fischer et al., 2021; Foster and Brozović, 2018; Geerts and Raes, 2009; He et al., 2022; Sapino et al., 2022; Smilovic et al., 2016). The function assumes a linear relationship between

relative crop yield losses and seasonal evaporative stress (deficits in crop water use) under water-limited climatic conditions:

$$1 - \frac{Y_w}{Y_p} = K_y\left(1 - \frac{ETa}{ETo}\right), \tag{9}$$

where $Y_w$ [ton ha$^{-1}$] and $Y_p$ [ton ha$^{-1}$] are water-limited and energy-limited potential yields (respectively), and $K_y$ [-] is the yield response factor that accounts for the complex crop characteristics that determine the crop water use. We redefined for AY, the ratio of water-limited yield $Y_w$, and potential yield $Y_p$ as a function of evaporative stress index $ESI = 1 - ETa/ETo$:

$$AY = 100(1 - K_y ESI). \tag{10}$$

The recommended $K_y$ values were set to 1.25 for maize, 0.9 for sorghum, and 1.15 for wheat, obtained from the FAO irrigation and drainage paper 66 (Steduto et al., 2012), and to 1.04 for teff (Araya et al., 2011).

### 2.3.4. Model evaluation

The CHC modelling framework was not explicitly calibrated, meaning all parameters were taken from literature and global dataset. Due to the limited availability of observed data we could not conduct a formal validation of the model. However, we compared the simulated values of the agrohydrological variables *Q, SM,* and *ETa*, as well as water-limited attainable yield *AY* with independent data. In our comparisons, we focused on the differences and correlations between the simulated and independent values using the root mean squared difference (RMSD) and the Pearson correlation coefficient ($\rho$).

The simulated mean annual *Q* was compared to published mean annual runoff observations from 20 sites within the RFA region (Table S3). We compared the annual cycles and interannual variabilities of the CHC-simulated SM and ETa with those of independent satellite- and model-based products (hereafter referred to as global datasets) for the period 2003-2010, across the four climatic regimes – semi-arid, dry sub-humid, sub-humid and humid. For AY, we compared the simulated AY with the ratio of Yw to Yp estimates obtained from published field trials, calibrated crop model simulations, and the GYGA database, at a total of 45 locations across the RFA region (Table S4). Additionally, we examined how temporal variations in AY explain variabilities in seasonal crop production. For this, we correlated AY and total cereal production (TCP) at each computation grid during the period 1995-2010.

### 2.4. Assessment of Green water availability and its yield potential

We evaluated the climatology of GWA and AY during the two growing seasons, Meher (May to September) and Belg (February to May) from the simulations for the reference period 1981-2010. We used soil moisture deficit (SMD) as a metric to evaluate GWA:

$$SMD = 100 \left(1 - \frac{\theta_{clim}}{\theta_{FC}}\right), \tag{11}$$

where $\theta_{clim}$ [m$^3$ m$^{-3}$] is the climatological mean seasonal soil moisture. *SMD* [%] is a dimensionless metric ranging from 0 (no moisture deficit) to 100 % (maximum moisture deficit). The yield metric, *AY* [%], which was determined using Eq. (10) is also a relative quantity that explains the percentage of the energy-limited potential yield that can be viably attained under the actual

water-limited conditions when all other agro-environmental factors such as nutrients are not limiting the yield, thus its values range from 0 (no yield) to 100 % (non-water-limited potential yield).

## 2.5. Future changes and climate sensitivity analysis

We applied the modelling framework to quantify changes in GWA and its implications for AY under climate change. We investigated the changes for three future periods: 2020-2049 (2030s), 2045-2074 (2060s), and 2070-2099 (2080s), under SSP1-2.6 (low), SSP2-4.5 (intermediate), and SSP5-8.5 (high) greenhouse gas emission scenarios. The impact assessments presented here are based on the median of downscaled multiple GCM projections of future changes in rainfall, air temperature, and solar radiation. Other climatic variables were assumed to remain unchanged (Peleg et al., 2019). The impact analyses compare SMD

and AY during the three future periods with the reference period (1981-2010) for the two growing seasons, Meher and Belg, and the four major crops grown in Ethiopia (teff, maize, sorghum, and wheat).

We also examined the sensitivity of AY to changes in rainfall (green water supply) and atmospheric evaporative demand (AED) as represented by ETo. We determined the rainfall- and AED-sensitivity metrics, $\beta_{RF} = \frac{\%\Delta AY_{(RF)}}{\%\Delta RF}$ [-] and $\beta_{ED} = \frac{\%\Delta AY_{(ED)}}{\%\Delta ETo}$ [-], as the ratio of percent change in $AY$ and percent change in rainfall or $ETo$ based on the one-at-a-time sensitivity

analysis approach (Hamby, 1994), where the CHC modelling framework was forced by the future RF and ETo (computed considering the future temperature and radiation) for all the three emission scenarios and three future periods. We determined the relative influences of the future changes in rainfall and AED on AY using the sensitivity ratio $\beta_{ratio} = \frac{\beta_{RF}}{\beta_{ED}}$. The values of $\beta_{ratio}$ [-] range from zero to infinity with values less than one indicating temperature sensitivity dominates and values greater than one indicating AED sensitivity dominates. This assumes that changes in rainfall and ETo are independent, which was

confirmed by the low dependency ($R^2 = 0.098$) between rainfall and $ETo$ computed for the period 2020-2099 for the three SSPs at every grid point.

## 3. Results

### 3.1. Evaluation of the CHC model

We first evaluate the simulated runoff ($Q$), Soil moisture (SM), actual evapotranspiration (ETa), and water-limited attainable

yield (AY) against other independent data. The comparison of simulated and observed mean annual $Q$ (Figure 3a) at the 20 field locations demonstrates a solid performance ($\rho = 0.91$ and RMSD = 52 mm) of the hydrological module in estimating the annual surface runoff from croplands. It is important to note that this comparison involves a point-to-grid value, and a perfect match cannot be expected due to this scale difference. The model underestimates the simulated $Q$ at the two most humid locations.

The comparison for SM is illustrated in the Figure 3b. The scatter plot of the area-averaged CHC-simulated annual SM (2003-2010) versus the median of that of the global SM datasets demonstrates that the simulated SM closely follows the interannual

variabilities in the median of the global datasets, with Pearson's ρ ranging from 0.69 in humid climate to 0.92 in semi-arid climate. Likewise, the annual cycles of the simulated and global SM datasets are also highly comparable, as shown in Figure S1 of the supplementary materials. However, the simulated SM estimates are (systematically) lower than those from the global SM datasets in all climatic regimes (see also Figure S3a and b). These differences decrease from semi-arid (RMSD = 0.075 $m^3$ $m^{-3}$) to humid areas (RMSD = 0.041 $m^3$ $m^{-3}$) as shown in Figure 4b. In semi-arid and dry sub-humid areas, these discrepancies can be partly attributed to the fact that satellite-based global SM datasets are informed by vegetation conditions in one way or another (e.g., Dorigo et al., 2017; Rodell et al., 2004; Teng and Parinussa, 2021), which carry soil moisture signatures from deeper soil layers compared to the top 60cm layer considered in this study. But in general, the differences observed in the present analysis is consistent with findings from a previous study by Jimma et al. (2023), which revealed that the global satellite-based soil moisture products (FLDAS, ERA5-Land, ESA CCI, and GLDAS2) significantly overestimate soil moisture over Ethiopia. Similarly, Teferi et al. (2023) also reported that soil moisture estimates derived from two satellite products – Advanced Scatterometer (ASCAT) and Soil Moisture Active Passive (SMAP), overestimate the insitu-measured soil moisture at a monitoring site in the Upper Blue Nile.

The simulated annual ETa correlates well with the median of the global ETa datasets in all climatic regimes with Pearson's ρ ranging from 0.80 in sub-humid to 0.93 in humid regions (Figure 3c). The annual cycles of the simulated ETa also closely match those of the global ETa datasets (Figure S2). However, during the dry months (December to February), the simulated ETa is lower compared to the global ETa across all climatic regimes. From the scatter plot presented in Figure 3c (see also the spatial patterns in Figure S3c and d), it is evident that the differences are more pronounced in semi-arid (RMSD = 61 mm) and dry sub-humid (31 mm) climates. Like in the case of SM, this can be attributed to the absence of vegetation representation in the CHC modelling framework, unlike in the global ETa datasets considered.

The comparisons of the simulated AY (at 45 locations) with three categories of the independent AY estimates are shown in Figure 3d. The green category (scatters) represents the simulated AY versus the corresponding independent AY estimates from field trials and calibrated crop modelling experiments at 20 locations in different years within the reference period (1981-2010) for various crops. These exhibited a good correlation (ρ = 0.79), and the smallest difference (RMSD = 7.1 %) compared to the other categories. The blue category represents the simulated AY versus AY estimates from field trials at 6 locations, conducted outside the reference period in later years (post-2010), which we plotted against the average simulated AY over the later years of the reference period (2007-2010). This category also showed a similar correlation to the green category but with higher differences (RMSD = 9.4 %). The orange category, representing the simulated versus GYGA-derived AY at 19 locations, showed a weaker correlation (ρ = 0.39) and the highest difference (RMSD = 14.3 %). Note that these independent data (both from field trials and model simulations) have their own uncertainties. For example, most of the field trials from which the paired Yw and Yp were derived were conducted to evaluate crop yields under supplementary irrigation, not explicitly to determine Yw and Yp. Therefore, the AY derived from these field trials are only estimates. Nevertheless, on average, the

CHC-simulated AY is still highly comparable to the independent AY data. The average simulated AY across the 45 locations

is 84.2 %, which is slightly lower than the average AY estimate from the independent data, which stands at 86.2 %.

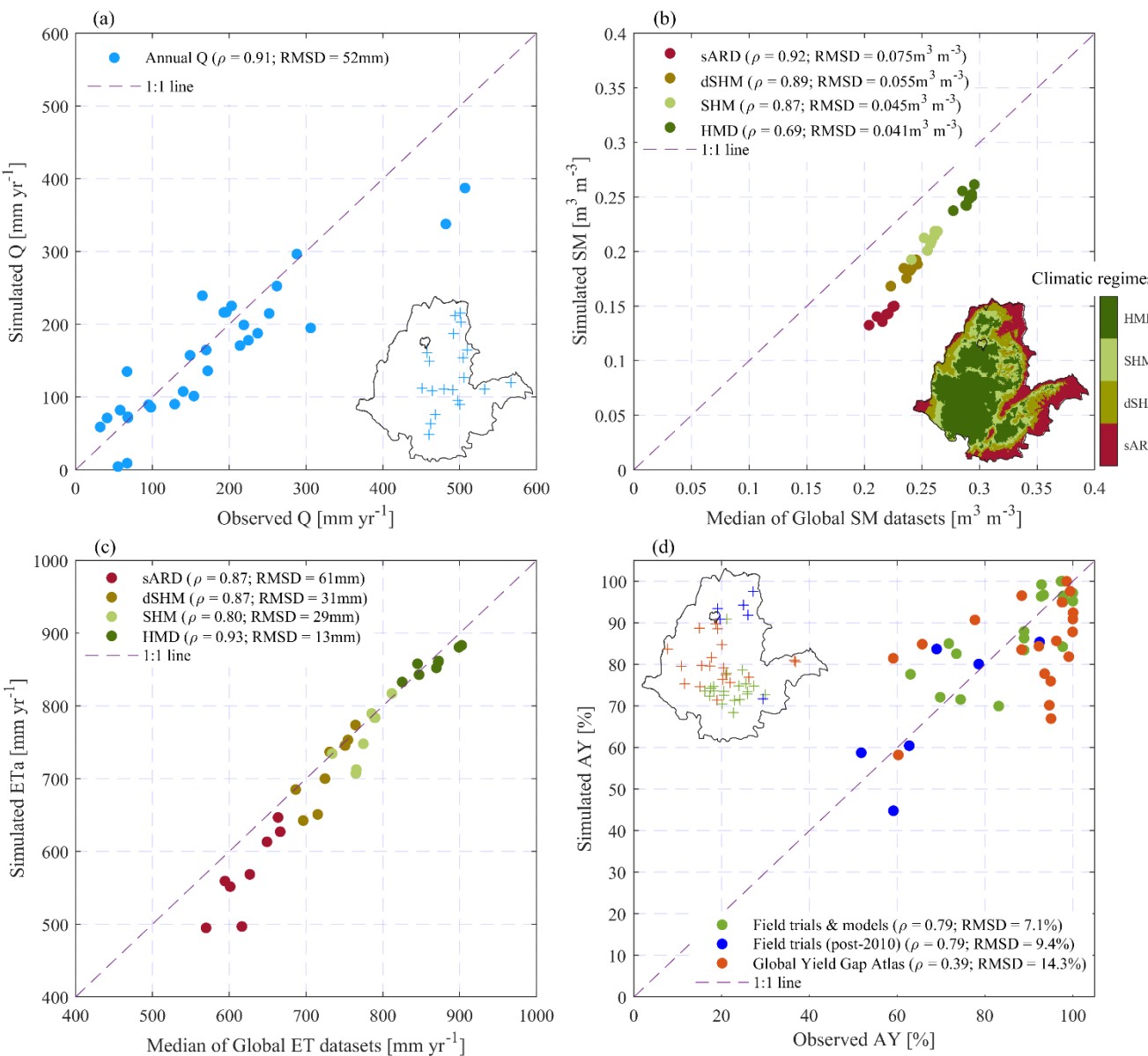

**Figure 3: Scatter plots of: a) Simulated versus observed yearly or mean annual runoff (Q), measured over varying years at 20 locations (shown on the inset map), collected from published studies across the RFA region (Table S3). This is a grid-to-point comparison. b) Simulated versus median of four satellite- and model-based annual (2003-2010) soil moisture (SM) products (Sect.**

**2.2) spatially averaged over the four climatic regimes – semi-arid (sARD), dry sub-humid (dSHM), sub-humid (SHM), and humid (HMD) -- shown by the inset map (from Figure 1). Each circle represents area-averaged (over each climatic regime) SM for each**

**year (2003-2010) and the colors indicate the climatic regimes. c) The same as in (b) but for actual evapotranspiration (ETa) considering the median of the five global datasets. d) Simulated water-limited attainable yield (AY) and the observed ratio of water-limited (rainfed) (Yw) to fully irrigated (Yp) collected published field trials and crop modelling experiments at 29 locations across**
**the RFA region indicated by on the inset map.**

We also evaluated the simulations of $AY$ [%] in terms of their correlation to variations in detrended total cereal production, $TCP$ [ton y$^{-1}$] as depicted in Figure 4. The rationale here is that in some regions TCP will follow the climate-driven attainable yield (high correlation), while in others the non-climatic factors, including nutrient input, improved cultivars, and pest and
weed management, will play a role (low or negative correlation). Because TCP in the RFA region of Ethiopia exhibited an average increase of 0.36 ton ha$^{-1}$ y$^{-1}$ during the studied period (1995-2010), despite an almost constant AY, we conducted the correlation analysis on detrended TCP, thereby removing some of the effects of the non-climatic component of the variabilities in TCP (Kukal and Irmak, 2018; Mohammadi et al., 2023; Wakjira et al., 2021). The correlation analysis between detrended TCP and AY anomaly reveals mostly positive correlations (median Pearson's $\rho = 0.37$), particularly in the primary Meher-
producing northern half of the RFA region (Figure 4).

Some areas, primarily in the humid and sub-humid regions of the southern and southwestern parts, and to a lesser extent in the western and southeastern parts of the RFA region, exhibited negative correlations between TCP and AY. This can be attributed to climatic regimes, land use, and socioeconomic practices. Two main factors may govern the AY-TCP correlation with climate. In less water-limited humid areas like the southwestern part, the linear relationship assumed in the crop yield response
to water availability (Eq. (9)) does not hold, as AY is weakly dependent on rainfall. Additionally, in these humid climates and areas with poor soil drainage (e.g., areas marked as E1 in Figure 4), crop yield can be adversely affected by saturated soil conditions, leading to waterlogging problems. Concerning land use and socioeconomic practices, the majority of areas with evident negative correlation are forest and/or agroforestry ecoregions (see E2 and E3 in Figure 4) (Kassawmar et al., 2018), while other areas are agropastoral (E4) where seasonal crop production is an optional practice. Thus, interannual variabilities
in TCP depend largely on how much farmers opt for cereal production in a given production year.

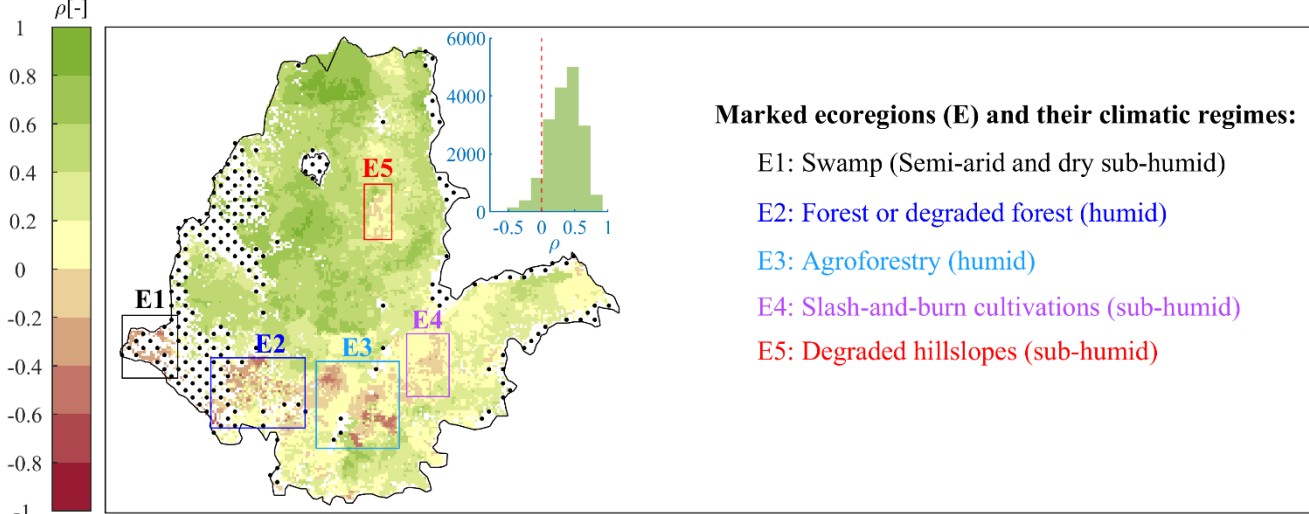

**Marked ecoregions (E) and their climatic regimes:**

E1: Swamp (Semi-arid and dry sub-humid)

E2: Forest or degraded forest (humid)

E3: Agroforestry (humid)

E4: Slash-and-burn cultivations (sub-humid)

E5: Degraded hillslopes (sub-humid)

**Figure 4: Map of Pearson's correlation coefficient (ρ) of detrended total cereal production (TCP) and AY for the period 1995-2010. TCP represents the sum of all cereals produced during the Meher growing season with average values ranging from 28 tons to 24,000 tons across the RFA region. The dots show uncultivated areas based on cropland cover fraction data from Copernicus Land Services (Buchhorn et al., 2020). The inset histogram shows the distribution of ρ. The rectangular marks (E1-E5) show ecoregions where the correlations are weak or negative.**

### 3.2. Green water availability and attainable crop yields

The reference climatology of growing season GWA and water-limited yield across the RFA region based on the computed SMD and AY values, is presented in Figure 5, considering the reference alfalfa grass (Ky = 1.1, Allen et al., 1998). During the Meher growing season, the southern and southwestern humid areas of the RFA region exhibit low soil moisture deficit, with values less than 10 % of $\theta_{FC}$ (see Figure 5a). Moving from the southwestern areas, the Meher SMD gradually increases in the northern and northeastern directions. In the peripheral semi-arid regions in the northeast, east, and southeast, the deficit reaches as high as 60-70 %. Notably, areas with SMD values below 20 % largely have AY>90 %. In other words, the water-limited yield gap of less than 10 % of the potential yield achievable under unstressed moisture conditions (equivalent to a fully irrigated system) can be attained in the south-central, southwestern, and eastern highland parts of the RFA region (Figure 5b). As expected, low AY (mostly less than 40 %) is evident in the southern and southeastern semi-arid parts of the RFA region during the Meher growing season. In summary, the median Meher SMD values are 17 %, 30 %, 37 %, and 53 % in humid, sub-humid, dry sub-humid, and semi-arid climates, respectively, while the corresponding median AY values are 93 %, 80 %, 68 % and 46 % (Figure 5e, f).

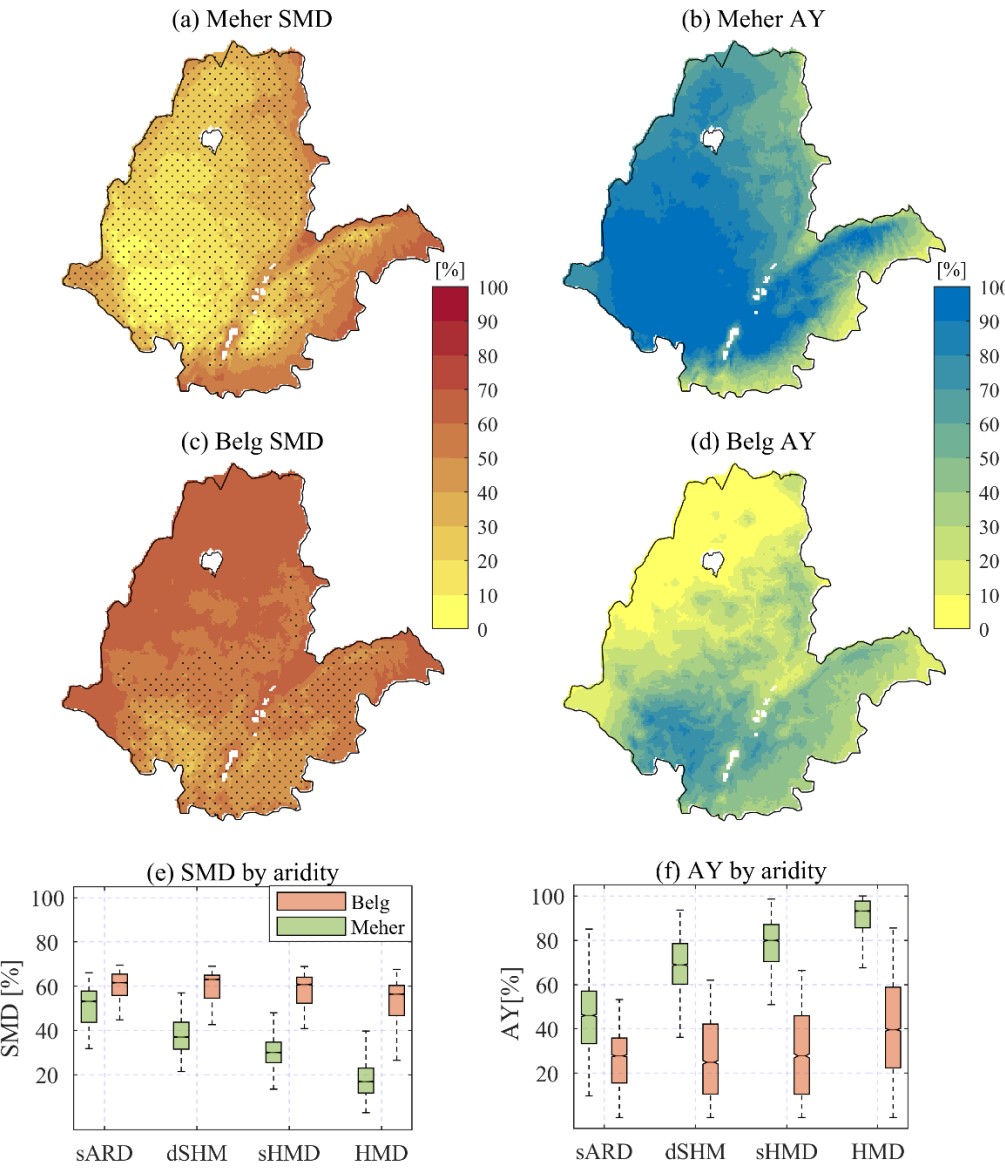

**Figure 5: Climatological SMD during Meher (a) and Belg (c), AY for alfalfa grass (Ky = 1.1) during Meher (b) and Belg (d), SMD in different climatic regimes by aridity (e), AY in different climatic regimes during the two growing seasons for the period 1981-**

**2010. Aridity classification is given in Figure 1: sARD = semi-arid, dSMD = dry sub-humid, sHMD = sub-humid, HMD = humid. The dotted areas in (a) and (c) show the Meher- and Belg-producing regions respectively, delineated based on the Atlas of Ethiopian Rural Socioeconomy (IFPRI and CSA, 2006).**

In the Belg season, the median SMD is mostly below 40 % in the major Belg-producing areas in the south (see Figure 5c), with AY of up to 80 % in the humid areas in the southwestern part (Figure 5d). In the central and eastern parts of the RFA

region, which constitute other Belg-producing regions, SMD is higher, reaching up to 60 %, resulting in a correspondingly

lower AY of 40-60 %. The northern and northwestern parts of the RFA are dry during the Belg season, thus SMD is very high. Unlike in the Meher season, the differences in SMD among the climatic regimes are less pronounced in the Belg, with median SMD ranging from 56 % in humid areas and 62 % in semi-arid regions (Figure 5e).

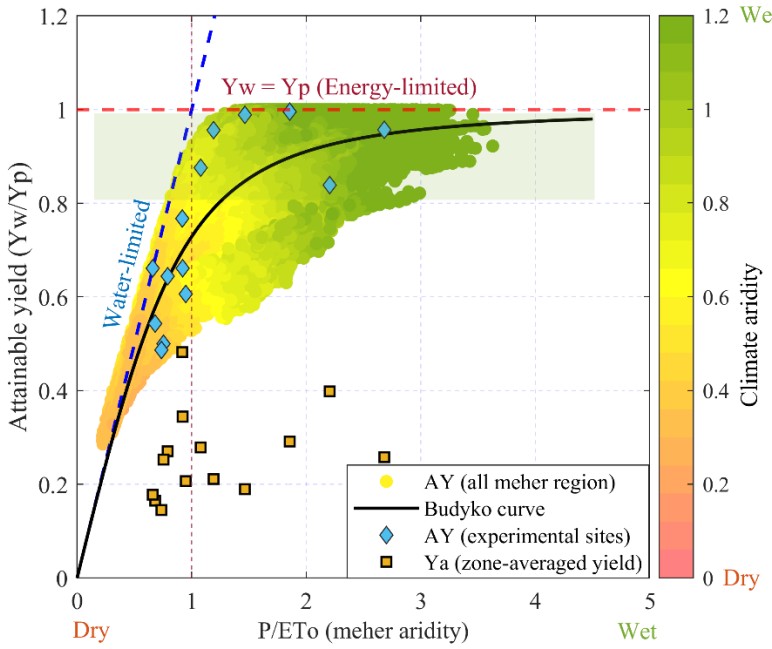

**Figure 6: Scatterplot of water-limited attainable maize yield fraction (Yw/Yp) against seasonal aridity showing the energy (dashed red line) and water limitations (dashed blue line). The color gradient shows the climatic aridity of each grid cell (different from the values on the x-axis, which are seasonal aridity for Meher). The solid line represents the parametric Budyko curve fitted to the point cloud. The blue diamonds show Yw at 14 locations across the RFA region, derived from published maize Yp data (fully irrigated, optimally fertilized). The orange squares are the corresponding average actual maize yield (Ya) in the administrative zone within which the experimental location is found. The shaded area indicates maize AY>80 %.**

In Figure 6, we presented attainable yield (Yw/Yp) in a Budyko-like space (Budyko, 1958) and included paired Yw (blue diamonds) and Ya (actual yield, orange squares) estimates at 14 locations from previous studies across the RFA region (Table S5) highlight the gaps between these two yield levels for maize. In the Meher season, green water has the potential to produce more than 80 % of Yp across 52 % of the Meher-producing regions of Ethiopia (highlighted by the shaded area in Figure 6). Only about 27 % of the Meher-producing region is moisture-limited (P/ETo < 1). Ya averages only 36 % of Yw at the 14 locations, indicating significant potential for yield improvement through enhanced management practices. Notably, there are areas where the seasonal rainfall is greater than the atmospheric evaporative demand, but AY is still low. This discrepancy explains the non-linearity of the relationship between AY and SMD (shown in Figure S4), which is primarily linked to the

combined effects of climatic and soil characteristics that influence rainfall-runoff partitioning and green water storage capacity. For example, a low AY for a given SMD can be associated with sandy soil in a dry climate, while a high AY for the same SMD can be linked to loam soil in a humid climate. From a management perspective, there may be limited options to address inherent soil physical properties like textural composition, apart from measures such as amendment with organic matter, which might to some extent, improve infiltration and green water storage capacity (Hartmann and Six, 2023).


### 3.3. Future changes in GWA and AY

### 3.3.1.     Projected changes in growing season rainfall and temperature

Based on the downscaled multiple GCM median projection, the growing season climate across the RFA region is expected to become warmer and wetter in future periods (Figure 7). The temperature increase is progressive over time, especially under

the intermediate- and high-emission scenarios, and shows little spatial variability (e.g., Figure 7c). However, changes in rainfall patterns vary across regions and depend on the growing seasons. For example, Meher-producing areas are likely to experience increased rainfall, while dry seasons are expected to remain unchanged or become even drier (e.g., Figure 7a, b). Similarly, during the Belg season, the Belg-producing areas in the southern part of the RFA region are expected to become wetter under higher greenhouse gas emissions. In general, a rainfall increase of up to 250 mm (multi-model median with 5-95 percentiles

uncertainty ranging from about -150 mm to +850 mm, not shown) is anticipated over a large part of the RFA region during the Meher season under the high-emission scenario in the 2060s (Figure 7a) and up to 300 mm in the 2080s (not shown). During the Belg season, rainfall increase could reach up to 150 mm (5-95 percentile of about -60 mm to 500 mm) under the high-emission scenario in the 2060s. Additionally, the mean annual temperature is projected to rise by up to 3 °C in the 2060s and up to 5 °C by the end of the century.

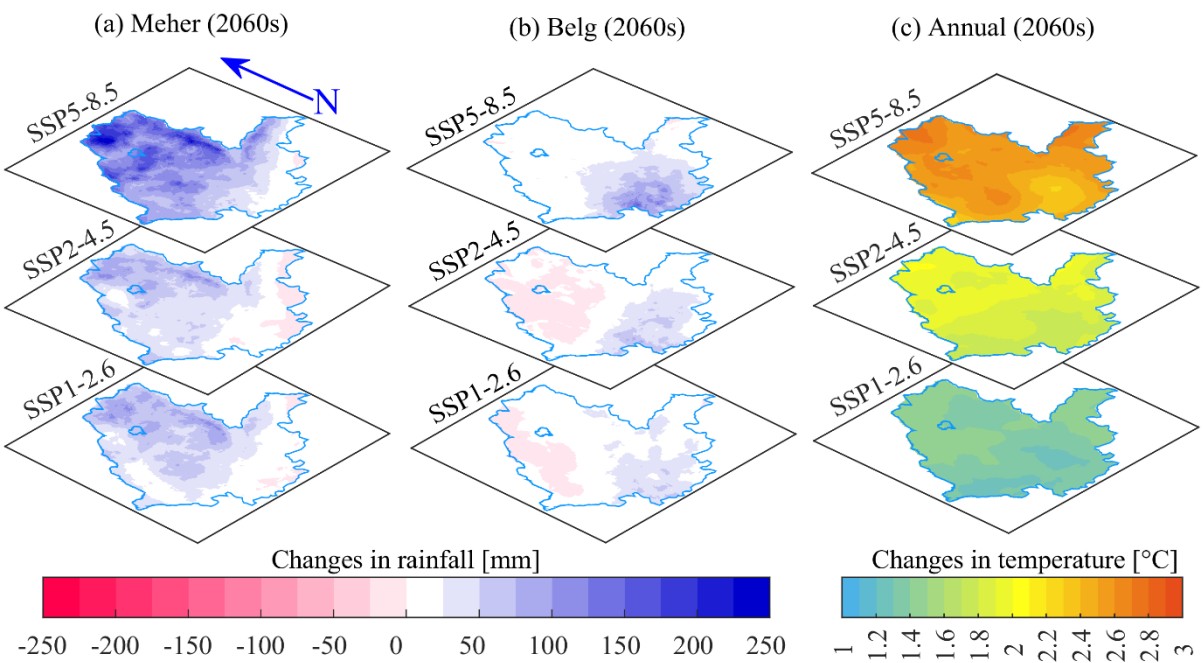

**Figure 7: Projected changes in seasonal rainfall in the 2060s during a) Meher (May-September), b) Belg (February-May), and c) changes in annual temperature under the three SSPs. The changes presented here are the median of 26 downscaled GCM projections for rainfall and 21 GCM projections for temperature.**

### 3.3.2. Future changes in soil moisture deficit

In the face of the expected warmer and mostly wetter climate across the RFA region of Ethiopia, the GWA is likely to increase. Here, we present the expected changes in GWA in Meher (Figure 8), as an estimated 88 % of the total grain production in Ethiopia occurs during this season. The expected changes in Belg are shown in Figure S5 of the supplementary materials. In Meher, SMD is expected to decrease by up to 5 % over a large part of the region in the 2030s under all emission scenarios (Figure 8). However, in the 2060s and 2080s, SMD is projected to increase by 3-15 % over the western, southwestern, and southern parts under the low and intermediate emission scenarios. This is expected, given the projected minimal change in rainfall under the warming climate over these regions.

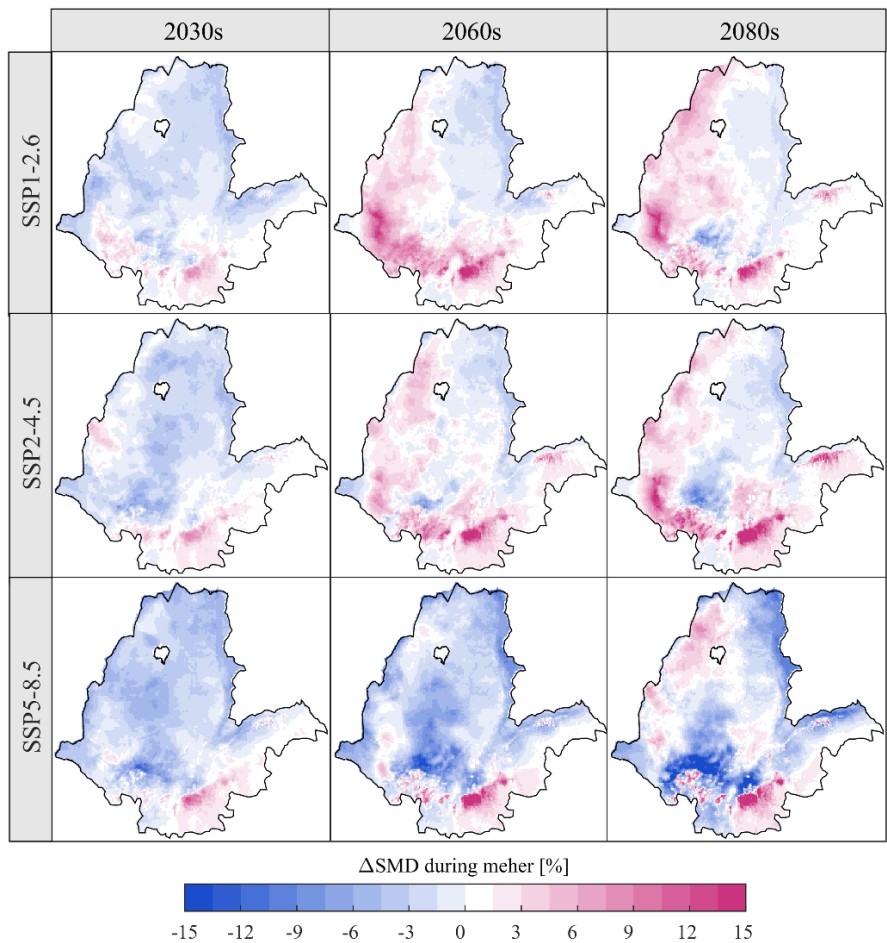

**Figure 8: Projected changes in soil moisture deficit (SMD) across the rainfed agricultural region of Ethiopia during the Meher growing season under the SSP1-2.6, SSP2-4.5, and SSP5-8.5 scenarios in the 2030s, 2060s, and 2080s.**

Under the high-emission scenario, soil moisture deficit is likely to decrease over the majority of the region, following the increase in rainfall. The central and northeastern parts of the RFA region are generally expected to experience a slight decrease in SMD and thus slightly higher GWA under all scenarios in the future. The highest increase in SMD was observed in the southwestern and southern parts of the RFA region, where the Meher rainfall mostly remains unchanged (see Figure 7a). In the Belg growing season, soil moisture deficit consistently decreases over the producing regions in the future periods under all

scenarios (Figure S5). These increases in GWA are particularly significant across the main Belg region in the southwestern part. In these areas, SMD is expected to decrease by up to 6 % in the 2030s (2020-2049) and by over 15 % in the 2080s under the low and intermediate emission scenarios. Under the high-emission scenario, GWA across the primary Belg-producing regions in the south and southwest is expected to increase by up to 23 % by the end of the century.

### 3.3.3. Future changes in water-limited crop yields

The likely implications of changes in SMD on attainable yields for the four major cereal crops (teff, maize, sorghum, wheat) cultivated in Ethiopia are presented next. We summarize the changes occurring during the two growing seasons in their respective producing regions (Wakjira et al., 2024) and across the various climatic regimes of the RFA region.

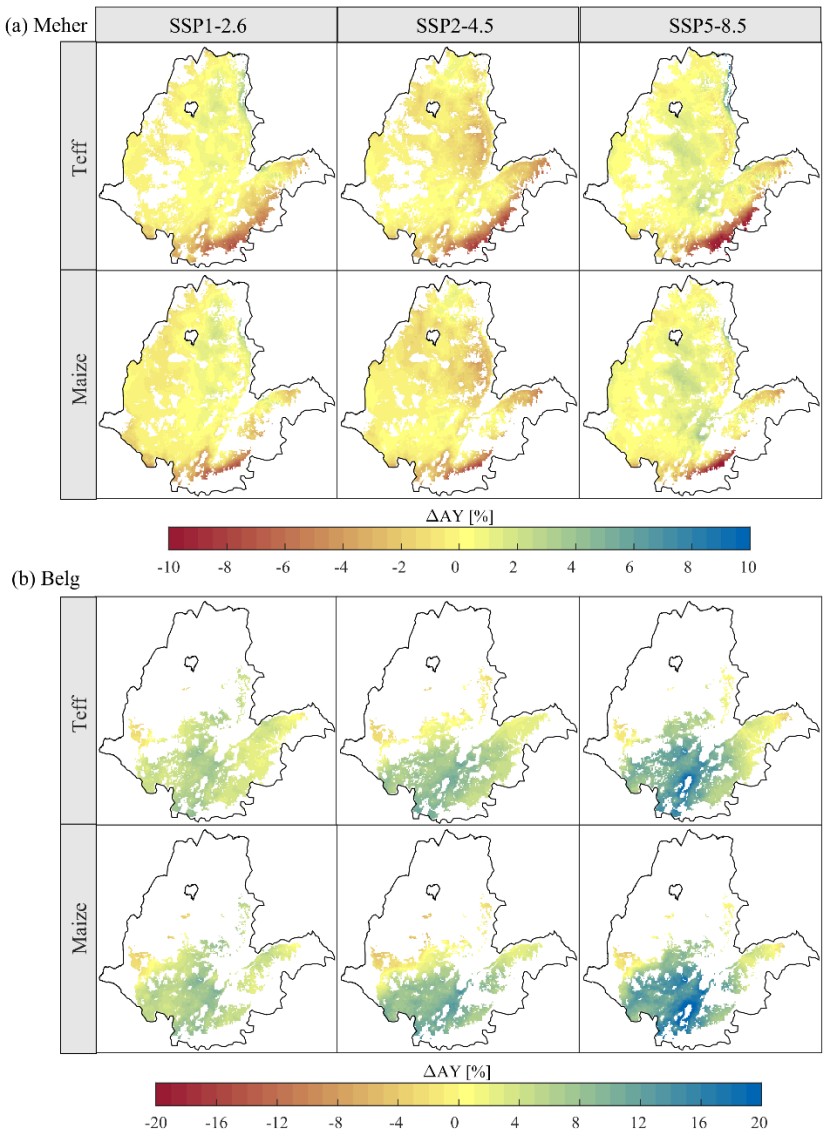

**Figure 9: Projected changes in water-limited attainable yield for teff and maize in Meher (a) and Belg (b) during the 2060s under the SSP1-2.6, SSP2-4.5, and SSP5-8.5. The RFA region was masked using cropland suitability maps (Wakjira et al., 2024) to restrict the analysis to areas potentially suitable for each crop. The non-producing areas during both seasons were also masked out following the Atlas of Ethiopian Rural Socioeconomy (IFPRI and CSA, 2006).**

**Meher season**

In the Meher season, minor or no changes in AY are expected, although regional differences exist in both the magnitude and direction of these changes. For example, for the 2060s, the projected changes in AY for teff range from -7.8 % to 5.3 % under the low-emission scenario, and from -12.1 % to 10.7% under the high-emission scenario. Similarly, for maize, the expected changes range from -7.8 % to 4.7 % under the low-emission scenario, and from -11.5 % to 6 % under the high-emission scenario (Figure 9a). Under the intermediate emission scenario, the changes are predominantly negative, ranging from -9.3 % to 1.8 % for teff and from -8.2 % to 1.2 % for maize. Similar order of magnitude and direction of changes were observed for sorghum and wheat crops (refer to Figure S6-9).

Comparing the near- and long-term future changes, we note a decreasing trend in AY for all crops under all emission scenarios. During the 2030s, the northern parts of the RFA region are likely to experience increases, while decreases are mainly evident in the southern and southeastern parts of the Meher-producing areas. Notably, the marginal areas in the south and southeast are likely to experience the most significant decrease in AY for all crops and under all scenarios. By the 2080s, most Meher-producing areas are expected to witness either no change or a decrease in AY (Figure S6-9).

We also examined the water-limited attainable yield responses in different climatic regimes (Figure 10). In humid areas, the temporal changes and spatial variability in AY are small for all crops under all emission scenarios. This is because these areas rarely have moisture limitations, meaning that an increase in rainfall will have a minimal effect on yield improvement. In contrast, in semi-arid, dry sub-humid, and sub-humid areas, AY is likely to increase in the 2030s and then consistently decrease in future periods under both low- and high-emission scenarios. Under the intermediate scenario, AY is expected to decrease in the 2060s and then increase in the 2080s. Spatial variations are particularly high for teff, especially in semi-arid areas with changes ranging from -10 % to +5 % in the 2060s.

**Belg season**

Following the increased rainfall and the subsequent rise in GWA, AY is projected to increase significantly and progressively over the major parts of the Belg-producing regions in future periods. However, a few areas will experience small decreases (Figure 9b). For teff, the expected changes in the 2060s range from -3.5 % to 8.8 % under the low-emission scenario and from -3.6 % to 21.7 % under the high-emission scenario. Similarly, for maize, the changes range from -5.9 % to 10.6 % under the low-emission scenario and from -4.7 % to 27.6 % under the high-emission scenario. These positive changes are expected to intensify, particularly under the high-emission scenario, with the majority of the Belg-producing region experiencing an increase in AY of over 20 % for all crops examined. The changes for all crops and future periods have been illustrated in the supplementary material appended to this paper (Figure S10-13).

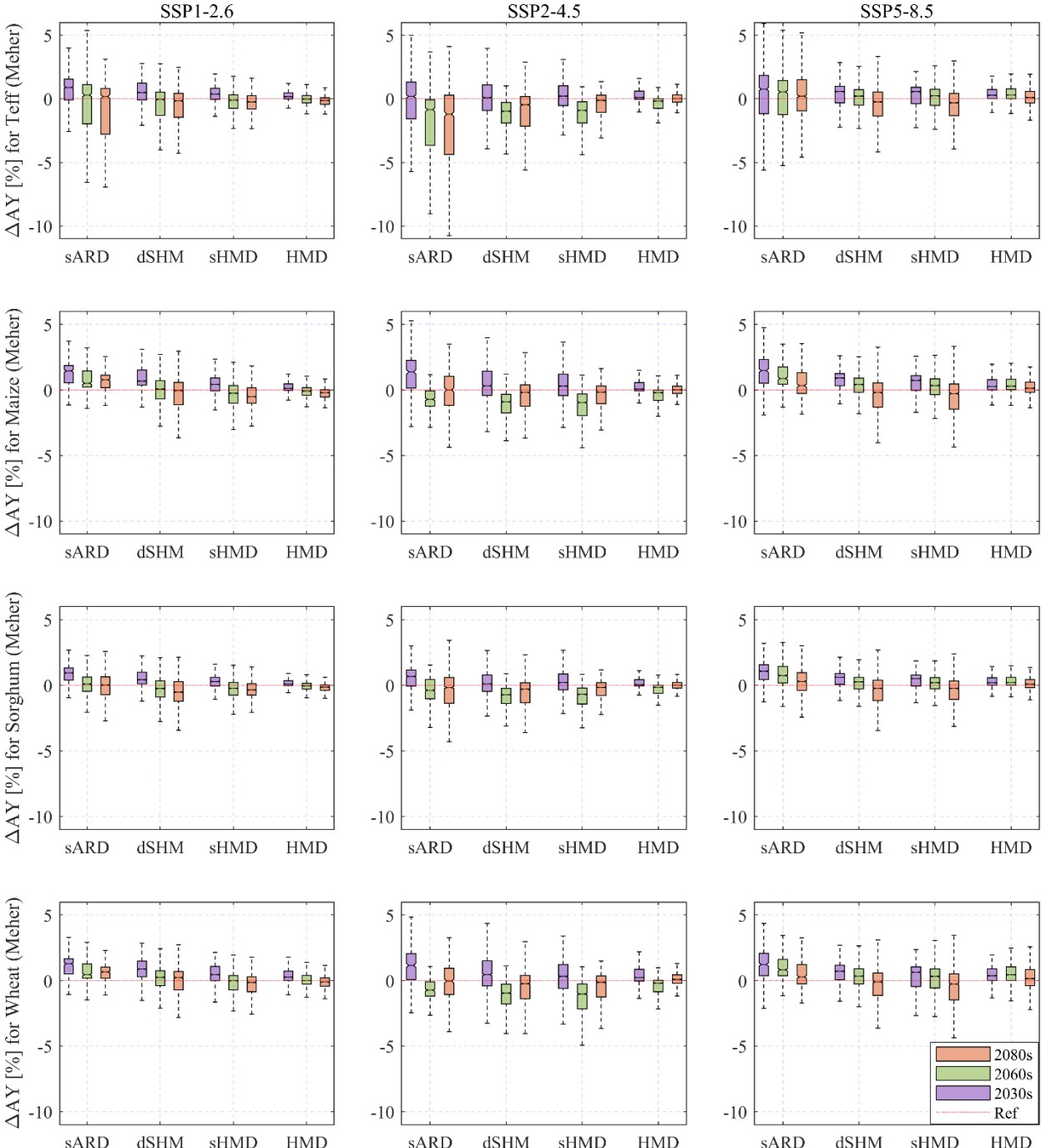

**Figure 10: Boxplots of the projected changes in water-limited yields (AY) of the four major cereal crops produced in Ethiopia in different climatic regimes under the three SSPs during the three future periods, during the Meher growing season. Each boxplot represents the distribution of AY changes within Meher-producing areas for all grid cells in the respective climatic regime. Outlier values have been excluded.**

The changes in AY during the Belg season in different climatic regimes are illustrated in Figure S14. The variations in overall changes in AY among these regimes depend on the emission scenario. Under the low-emission scenario, median changes in AY do not significantly vary across climatic regimes, remaining positive throughout the three future periods for all crops examined. In contrast, under the intermediate and high-emissions, AY tends to increase with climatic wetness (from semi-arid to humid climates). Decreases in AY are mainly noticeable in semi-arid and dry sub-humid areas in the 2030s, particularly under the intermediate emission scenario. However, during the 2060s and 2080s, median changes in AY are positive and relatively consistent across climatic regimes for all crops. The projected increases in AY are higher for maize and wheat compared to teff and sorghum. Spatial variability in changes generally increases with climatic wetness and into the future periods under the low-emission scenario while a decreasing pattern rarely observed under the intermediate and high-emission scenarios.

### 3.3.4. Climate sensitivity of attainable crop yields

The spatiotemporal average relative sensitivity ($\beta_{ratio}$) of AY to atmospheric evaporative demand (AED) and rainfall for the four crops under the three SSPs are presented in Figure 11. The results are averaged for the three future periods for the main growing season Meher at the administrative agricultural zones of Ethiopia. Results for the Belg short growing season are presented in Figure S15. The climate sensitivity is reported at a zonal scale, rather than at the grid scale, to provide zone-specific insights into the relative importance of ETo and rainfall in determining future green water productivity; this information is crucial for agricultural water management planning, service provision, and decision-making at the institutional scale. For the interpretation, we categorize the sensitivity of AY as AED-sensitive ($\beta_{ratio} \leq 0.8$), AED- and rainfall-sensitive ($0.8 < \beta_{ratio} < 1.2$), and rainfall-sensitive ($\beta_{ratio} \geq 1.2$).

During the Meher growing season, AY is predominantly AED-sensitive for all crops under all emission scenarios we considered (Figure 11). The percentage of zones across which AY is, on average, AED-sensitive is 64 % for teff, 50 % for maize, 52 % for sorghum and 77 % for wheat under low emissions. The influences of AED on AY increase with GHG emissions. Under high emissions, AED dominates the changes in AY in 82 %, 65 %, 73 %, and 92 % of the zones for teff, maize, sorghum, and wheat, respectively. In the remaining zones, the simultaneous influences of rainfall and AED are evident, except for maize, where rainfall has a more significant impact than AED. Notably, changes in rainfall have a low influence on teff and wheat, with AY being rainfall-sensitive in less than 5 % of the zones. These crops are exceptionally more sensitive to

AED compared to maize and sorghum, as indicated earlier. Temporal changes in climate sensitivity of AY are also evident, but the spatial patterns do not change significantly in future periods.

The influences of changes in AED and rainfall on AY are primarily linked to the climatic regimes of the RFA region. In semi-arid and dry sub-humid climates, AY is predominantly AED-sensitive. This is particularly noticeable in the northeastern zones (covering the eastern parts of Tigray and Amhara, and western zones of Afar regions) and the central to eastern zones (including East Shoa, Arsi, and Hararge zones of the Oromia region) in the RFA region for all crops under low emissions (left panels in Figure 11). This influence of AED intensifies to more zones in sub-humid and humid climates under the intermediate and high
emissions. Rainfall-sensitivity is largely evident in humid climates, mostly in the central, western, and northwestern zones under low emissions for all crops except wheat, which is rainfall-sensitive only in the northwestern parts of the RFA region (Western Tigray and North Gonder zones). Maize AY is strongly rainfall-sensitive under all scenarios, across the humid zones in the western part of Oromia (Wollega, Jimma, and Illubabor zones), Sidama region, and most of the zones of the SNNP region in the southwestern part.

In contrast to the main growing season, future changes in AY during the Belg growing season are primarily influenced by changes in rainfall, particularly in the main Belg-producing areas in the southwestern, southern, and southeastern parts of the RFA region for all crops under all SSPs (Figure S15). There is less spatiotemporal change in the rainfall sensitivity of Belg AY. Only under the high-emission scenario does the influence of changes in AED contribute to the influences of rainfall, primarily in the northern half of the RFA region, which remains dry during this season.

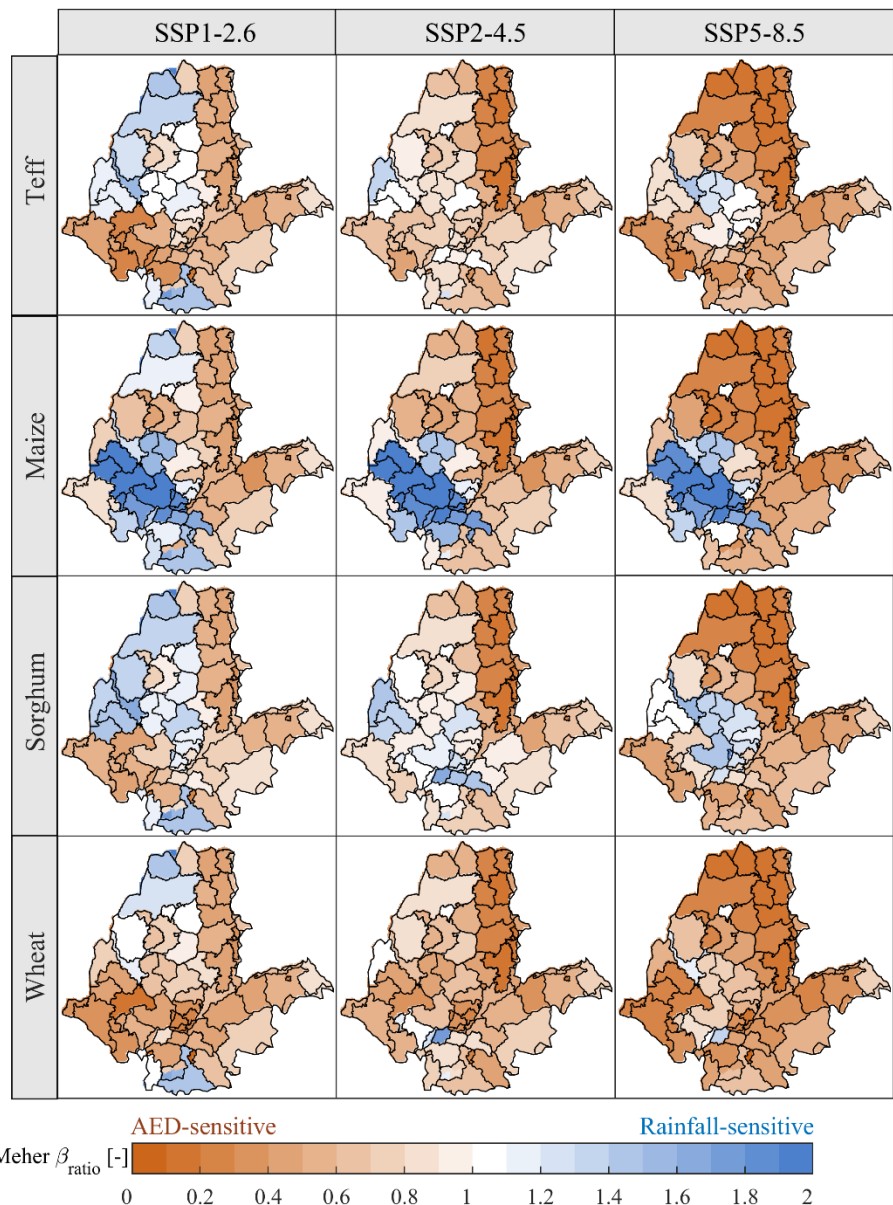


**Figure 11: Area-averaged relative sensitivity ($\beta_{ratio}$) of water-limited attainable yields (AY) to rainfall and atmospheric evaporative demand (AED) for the Meher growing season at the administrative zone level under the low-, intermediate-, and high-emission scenarios. The mapped values represent the average of $\beta_{ratio}$ of all grid cells within each zone, and all three future periods. The names of the administrative zones are indicated in Figure 1 and Table S1.**


## 4. Discussion

### 4.1. Future green water limitations and crop yields in Ethiopia

Achieving crop yield at its potential level is not only a matter of adaptation to climate variability and change, but it is also a key strategy for eradicating rural poverty and building resilience in heavily agricultural nations like Ethiopia (Abrams, 2018;

Tittonell and Giller, 2013). In RFA systems green water is a major limiting factor, and thus, Yw defines the rainfed crop yield potential (Dijk et al., 2017; van Ittersum et al., 2013). Our analysis sheds a broader light on the potential of green water for crop production, its spatial variability, and temporal changes (at climatological timescale) in the face of climate change across the RFA region of Ethiopia.

We showed that the likely wetting and warming climate across the large part of the Ethiopian RFA region is expected to

slightly enhance green water availability (GWA) in the near future. This is largely consistent with a recent global-scale assessment by Liu et al. (2022), which reported a decrease in agricultural water scarcity of up to 15 % in the RFA region for a comparable future period (2026-2050) and reference period (1981-2005) under the low-emission scenario. However, it is also evident that in the long term, the effects of climate warming outweigh those of climate wetting, leading to green water scarcity. For example, Setegn et al. (2011) in a SWAT-based evaluation of future agricultural water availability in the Lake Tana basin

in northern Ethiopia using four GCMs, revealed a decreasing tendency in soil moisture (by up to -2 %) during the mid and end of the century, which aligns with our estimates for that area (-3 % to +1.5 %, Figure 8). In the northeastern semi-arid areas, GWA is expected to improve under all emission scenarios, across all future periods, and in both growing seasons. This is consistent (in terms of the direction of change) with another recent assessment based on the SWAT+ model driven by multiple GCM projections under medium- and high-emission scenarios, which reported substantial increases in GWA in future periods

in the Kobo-Golina River catchment (Abate et al., 2024). Similarly, our findings in the central Rift Valley support those of Muluneh, (2020), who reported increases in soil moisture based on AquaCrop model for maize forced by ensemble GCM projections under the high-emission scenario. The observed decreases in GWA across the southeastern semi-arid areas is the combined effects the no change or drying tendency, and temperature warming. Our findings in this area contradicts with those of Serur, (2020), which predicts increases in GWA in Weyn River catchment.

It has been widely reported that crop yield across the tropics, including East Africa, are likely affected negatively by climate change (e.g., Asseng et al., 2015; Jägermeyr et al., 2021; Li et al., 2022; Rosenzweig et al., 2014). We observe overall decreasing tendencies in Meher season AY for the four crops studied, across all climatic regimes, under all emission scenarios by the mid and end of the century (Figure 10). In this regard, our results align with a recent study (Yang et al., 2023), which assessed climate-driven changes in maize yields in Ethiopia using the process-based DSSAT model with four GCM projections

and similarly indicated a decreasing trend in yields. Our findings also align with the work of Rettie et al. (2022), which used an ensemble of crop models and similarly concluded decreases in Meher maize and wheat yields by mid-century. Consistent with our findings (Figure 11), they also reported that maize and wheat yields are more sensitive to temperature changes than

to changes in rainfall. In Gambella, western Ethiopia, Degife et al. (2021) observed a decreasing trend in maize yields throughout the 21st century, with temperature playing a strong role in driving the change, which aligns with our findings except for the 2030s, for which we found no change or a slight increase in that region (Figure S8). Additionally, the projected slight increases in maize AY in Bako (central west) and slight decreases in Melkasa (central east) are comparable to those of Araya et al. (2015). However, our findings largely contradict the crop yield projections based on a statistical model forced by two regional climate model projections (Kassaye et al., 2021), which reported substantial increases in teff and maize yields and substantial decreases for sorghum and wheat.

## 4.2. Uncertainties and limitations

From the model evaluation (Sect. 3.1), it is evident that the CHC modelling framework can reliably estimate the agrohydrological variables at a grid scale with minimal complexity for our stated purpose, which is to inform long-term agricultural water management, climate adaptation, resilience, and rural economic development in Ethiopia. Before we further discuss the practical implications of the GWA and AY conditions across the RFA region, we first highlight the limitations of the CHC model structure, especially the considerations needed when applying the model in different contexts, particularly in terms of purpose and temporal scale.

One of the simplifications in the CHC modelling framework is the use of the CN-based soil water balance model. The main limitation of the method stems from the empirical parameters – namely the curve number, and initial abstraction ratio, used to simulate the vertical soil water balance at grid scale without routing the surface as well as subsurface flow. Therefore, these parameters must be properly calibrated for site-specific conditions (Assaye et al., 2021; Qi et al., 2020), particularly when the model is applied at local scale and shorter time scale. In our application, calibration over the entire RFA region is rather difficult as this would require large local observation data.

We quantified the uncertainty associated with the CN-based rainfall-runoff partitioning approach (CHC-CN) by comparing it to a water balance model that is based on the maximum soil infiltration capacity (CHC-WB), another widely applied concept (e.g., Chiarelli et al., 2020; Hoogeveen et al., 2015; Sishu et al., 2024; Tenreiro et al., 2020). In CHC-WB we assumed that surface runoff events occur when rainfall on a given day exceeds the maximum infiltration capacity (the difference between water content at saturation, and soil water content on the day before). Comparing the AY simulated using CHC-CN and CHC-WB, we found negligible differences between the models (Figure 12), with root mean squared deviation (RMSD) from the observed AY of 10.98 % for CHC-CN and 10.86 % for CHC-WB. Similarly, the differences in ETa and SMD simulated by the two models were small (see Figure S16b and c). The main discrepancy appeared in surface runoff (Q) and deep percolation (Dp) simulations (Figure S16a), as the CHC-WB model does not adequately differentiate surface and subsurface flow, diverting a larger portion of runoff to deep percolation. Although both surface runoff (Q) and deep percolation (Dp) are not the primary variables of interest in our analysis, their accurate estimation is crucial for other agrohydrological applications,

such as water harvesting, flood (spate) irrigation design, and agricultural drainage system design, and monitoring soil and nutrient erosion. When input parameters like rainfall intensity and infiltration characteristics are available, and the model's complexity is manageable, infiltration-based models, such as those using Richards' equation (Richards, 1931) may provide more precise results than the curve number and water balance-based models.

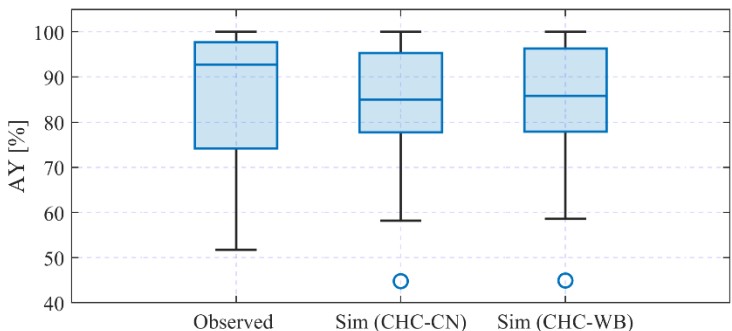

**Figure 12: Comparison of CHC-simulated AY using the CN-based and water balance (WB)-based rainfall-runoff partitioning methods at 45 locations (Table S4) for which independent (observed) AY were collected published field trials, crop model simulations and GYGA ([www.yieldgap.org](www.yieldgap.org)).**

The crop module of the CHC modelling framework estimates relative crop yield only at a seasonal scale using the evaporative stress index (ESI) as a climatic factor and the yield response factor (Ky) to represent the seasonal crop response. In crop and
agroecosystem modelling, such models are categorized as crop coefficient models and are primarily used for planning and decision-making (Foster and Brozović, 2018; He et al., 2022; Schwartz et al., 2020). Sub-seasonal variabilities of stress conditions (water, thermal, light, nutrient, etc) as well as the crop physiological responses within the growing season have not been represented in the CHC modelling framework. Therefore, the use of the framework should not aim for accurate simulation of crop growth and yields; such simulations should be based on process-based crop models, which are capable of accounting
for major influences of climate change such as the negative effects of heat stress and the positive effects of elevated $CO_2$ on future crop yield (Becker et al., 2023; Lobell et al., 2011b). In our analysis, which focuses on assessing changes on a climatological timescale, we focused on the rainy seasons (May to September, and February to May) climate and instead of accurately defining the growing period from planting to harvest. While the uncertainties associated with the choice of growing period are averaged out at climatological times scale, accurate planting date is an important input in process-based crop growth
and yield simulations at local and sub-seasonal scales (Lala et al., 2021).

Despite the uncertainties and limitations outlined, the results presented are robust and offer valuable insights for practitioners in planning and policy formulation in Ethiopia. Furthermore, the model developed can be adapted for application in different regions and climates, enhancing its utility for broader decision-making processes.

### 4.3. Implications for green water management and sustainable intensification

While the changes in water-limited attainable yield vary greatly across the RFA region in terms of both magnitude and direction during the main growing season, the results show that semi-arid and dry sub-humid areas are most likely to experience a reduction in AY. On one hand, this is attributed to an intensified soil moisture deficit, as evident for example in the northwestern, western, southern, and southeastern semi-arid areas (with low and intermediate emissions in the 2060s and 2080s, in Figure 8), and on the other hand, it is also a result of increases in AED-sensitivity (Figure 11), particularly in the

moisture-limited regions. In these regions, it would be beneficial to implement on-farm water management strategies that maximize green water availability and minimize non-productive green water flows.

Practices that aim to maximize GWA should focus on altering the rainfall-runoff partitioning processes by increasing the opportunity for infiltration during rainfall through various surface management practices. For example, tillage and physical measures like bunds, infiltration trenches, tied-ridge, and planting pits among others (Hurni, 2016; Makurira et al., 2009;

Nyakudya et al., 2014) have been successfully evaluated in field experiments and on-farm practices. Other measures like residue retention and cover cropping not only enhance infiltration, but also suppress the non-productive evaporation from the soil surface (Rockström, 2003). Additionally, measures that improve infiltration also offer the side benefit of reducing soil erosion by runoff, another critical challenge that contributes to the crop yield gap. Finally, we strongly suggest that the selection of water management practices should be carefully made by evaluating their need and suitability primarily based on climate

and soil characteristics. For example, in humid climates with heavy clay soils, such practices may result in waterlogging problems, which is also a major yield-reducing factor in such environments (Manik et al., 2019; Pittelkow et al., 2015).

The projected increase in GWA and AY during the Belg season may provide an additional opportunity for farmers to intensify their production during this season. This, however, will need firm stakeholder commitment to plan and mobilize resources for action in the framework of National Adaptation Plans (NAP) and similar initiatives (Conway and Vincent, 2021). Long-term

awareness of stakeholders, ranging from the institutional level to farm-level actors, on the expected challenges and opportunities of climate change, supported by climate information services for short-term decisions, is highly important to exploit such opportunities (Grossi and Dinku, 2022). In addition to climate information like the forecast information of the expected onset and cessation of Belg rain, extensive support on proper selection of crop types compatible with the duration of the short growing season is vital to help farmers effectively plan and undertake bi-annual production without compromising

the main growing season.

### 5. Conclusions

We investigated the cascading effects of climate on green water availability and water-limited attainable yield (AY) in the context of the rainfed agricultural region of Ethiopia. We integrated hydroclimatic processes with crop yield response through an agrohydrological modelling framework to assess the current potential, future changes, and climate sensitivity of AY. The

AY across the Ethiopian RFA region is 79 % of what could be produced under water-unlimited conditions during the main

growing season (Meher) and 37 % during the shorter season (Belg) during the reference period (1981-2010), with regional variation depending on the climatic regimes. The soil moisture deficit (percent of soil moisture content at field capacity) during this period is on average about 29 % in Meher and 56 % in Belg.

The future climate over the RFA region is expected to be warmer and mostly wetter. Under these changes, future changes in green water availability and AY vary across regions, emission scenarios, future periods, and between the two growing seasons. In Meher, the expected changes in AY predominantly fall within the ±5 % range under all scenarios and future periods. Changes during the 2030s are largely positive under all scenarios, but AY shows overall decreases in the 2060s and 2080s. Decreases in AY are mostly evident in semi-arid regions, with teff being the most affected. These changes are dominantly driven by the atmospheric evaporative demand (AED), that is caused by temperature increase, especially in moisture-limited regions. The influence of AED increases under the intermediate and high-emission scenarios, suggesting the need for due attention to management strategies that suppress evaporative losses in the future. In Belg, AY is expected to progressively increase by up to 20 % under the high-emission scenario by the end of the century, providing an opportunity for farmers to expand crop production in this season. These changes are dominantly driven by increases in rainfall, implying that green water management practices that increase water availability would further improve crop yields during this season. Furthermore, our assessment of documented field experiment results and crop yield data from the RFA region reveals a large gap between the actual and water-limited yield. For example, for maize on average only 36 % of AY is actually realized under the current practices, suggesting that green water management practices should be combined with other measures that overcome the yield-reducing factors related to soil nutrient, tillage practices, plant protection, and cultivar improvement.

Finally, the CHC modelling framework developed in this study can be applied to conduct similar assessments in other regions. It can also be adopted for various agrometeorological applications, such as estimating seasonal water availability and crop water demand for both irrigated and rainfed systems, as well as predicting relative yield for management planning and decision-making. The framework minimizes dependence on process-based crop models for such analyses, which often require intensive measurement data for the calibration of several model parameters.

**Data availability**: The processed data and simulation output will be made available upon request.

**Author contribution**: Conceptualization: MTW; Data curation: MTW; Modelling and analysis: MTW; Methodological and result discussions: MTW, NP, JS, PM; Draft preparation: MTW; Revision: NP, JS, PM; Supervision: PM

**Competing interests:** At least one of the (co-)authors is a member of the editorial board of Hydrology and Earth System Sciences. The authors have no other competing interests.

**Acknowledgments**

We thank Silvan Ragettli for his valuable comments on the evapotranspiration data and outputs, and Moritz Laub for his insightful suggestions on the choice of suitable pedotransfer function. MTW was supported by ETH for Development (ETH4D), ETH Zurich, through the E4D Doctoral Scholarship Program. NP was supported by the Swiss National Science Foundation (SNSF), Grant 194649 ("Rainfall and floods in future cities").

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
