# Peer review of "Green water availability and water-limited crop yields under a changing climate in Ethiopia"

_Hydrology and Earth System Sciences, 2024_

## Author Comment (AC1)

**Response to Anonymous Referee #1**

First and foremost, we would like to thank the referee for their time devoted to reviewing our work and providing their comments and suggestions that are helpful for improving our manuscript. We will use these comments and suggestions in the revision of the manuscript. In particular, we will add a brief section to elucidate the key limitations of the modelling framework and provide recommendations on the applications of the framework for simulations in different spatiotemporal contexts.

Here, we will respond (in blue font) to the comments made by the referee (in black) point by point.

**General Comment**

The study focuses on the effects of climate change on green water availability and water-limited attainable yields (AY) for major cereal crops in Ethiopia. It develops an agro hydrological Modelling framework to simulate climatic-hydrological-crop interactions for the reference year (1981-2010) and future periods (2020-2099) under different greenhouse gas emission scenarios. The study discuss the importance of green water management practices to improve crop yields, especially in rain fed agricultural systems. It identifies a significant gap between actual and water-limited yields, emphasizing the need for integrated management strategies to overcome yield-reducing factors. The findings suggest that future changes in AY vary across regions, emission scenarios, and growing seasons, with temperature increase playing a key role in driving changes. The study concludes by recommending the adoption of climate-smart agricultural practices and the use of agro hydrological Modelling for informed decision-making in agricultural management planning. It is well written and have a significant contribution in green water management beyond the study area.

We thank the referee very much for the positive feedback on our manuscript.

The modelling approach used in this study provides valuable insights about the future without using process-based modelling. However, the approach is subjected to uncertainties and assumptions that may impact the accuracy of the results. Can the authors clarify as limitation or future research considerations in the following issues?

We thank the referee for the comment. Before we go into the specific comments below, we would like to clarify the context of this work. In our modelling approach, we emphasized capturing cascades of climatic, hydrological, and crop yield (in relative terms) information at *a climatological* time scale (average conditions over a 30-year period) at a spatial scale of *5 km x 5 km*. We aimed to provide a bigger picture that could be valuable for long-term agricultural water management planning and policymaking at national and sub-national scales, supporting effective climate adaptation, resilience, and rural economic development in Ethiopia. In doing so, it was necessary to make compromises between the *data* used, *models* employed, *geographical area* covered, and the targeted *information*. It

is expected that the uncertainties in data, parameters, as well as modelling concepts, unavoidably propagate to the outcomes we presented in the paper. We will briefly elaborate on these uncertainties in the revision.

- The partitioning of rainfall to runoff is based on the CN approach which is completely empirical and needs locally adjusted CN based on soil, land use and hydrologic conditions. How is this affecting the competition where there are no locally contextualized CN values?

  We acknowledge the uncertainties that could arise from the empirical nature of the CN-based rainfall partitioning method implemented in our modelling framework. As the referee rightly stated, the best way to reduce such uncertainty is to adjust the CN value based on the local context. This is feasible for assessments in specific locations (e.g., fields, farms, landscape scales) and small catchments. However, in our case, it is hardly practicable as the model was implemented at a grid-scale covering a significantly large geographical area (the whole agricultural region of the country, ~667,000 km²), which ideally requires adjusting the CN at every grid, preferably based on observed local soil and surface conditions.

  Instead, we adjusted the CN at every grid, considering: *land use* (we used CN corresponding to agricultural land from the USDA lookup table), *soil conditions* (we assigned the CN to each grid based on the Hydrologic Soil Group defining the soil infiltration characterisitics from Ross et al. (2018)), and *soil mositure conditions* (we updated the CN values daily in the soil water balance model using Equation 6 – on page 7). Another most common practice, especially in catchment hydrological modelling is to optimize the CN value through model calibration with observed surface runoff or stream flow data measured at a designated outlet point (e.g., Arnold et al., 2012; Qi et al., 2020). In our case, we focused on the vertical soil water balance without routing the surface as well as subsurface flow, thus this option does not apply in this context.

  Several hydrological models such as SWAT (Arnold et al., 2012), AGNPS (Young et al., 1989), HEC-HMS (USACE, 2000), etc also use CN methods as one option for infiltration estimation. Additionally, there is scientific interest in deriving high resolution CN estimates that can be combined with state-of-the-art landcover maps from remote sensing (e.g., Jaafar et al., 2019), which increases applicability of this method over a large geographical domain where calibration of this parameter is a challenge.

- What about using other partitioning approaches such as Thornthwaite and Mather soil moisture water balance model widely applied in the area?

  We believe different models are intended primarily for a specific purpose (although they can also be adaptable for other related purposes), and their parameterization, simulation time steps, and other features are defined based on that main purpose. In this regard, Thornthwaite and Mather is a monthly soil moisture and groundwater recharge model (Sishu et al., 2024; Steenhuis and Molen, 1986). It does not account for soil properties and land surface conditions that determine the rainfall-runoff processes, it is informed only by rainfall and potential evapotranspiration conditions (Sishu et al., 2024). In the version modified by the US

Geological Survey (Westenbroek et al., 2010), the surface runoff routine of the model is based on the CN method. We did not compare this model with our modelling framework, but we can say that a monthly model is too coarse to capture the agro-hydrological fluxes that are pertinent to plant growth response. Our agrohydrological modelling framework determines the soil water balance at a daily time step. This is particularly important to better capture the effects of weather-driven daily variations in antecedent soil moisture conditions that determine the partitioning of rainfall into infiltration and surface runoff. In fact, the soil water balance modules of many crop growth and agroecosystem models, such as DSSAT (Jones et al., 2003), AquaCrop (Hsiao et al., 2009; Raes et al., 2009; Steduto et al., 2009), SIMPLE (Zhao et al., 2019), and SWAT+ (Čerkasova et al., 2023), are built on CN-based approaches.

- Can authors clarify how the lateral and subsurface flow likely affect the uncertainty? This kind of flow is very important in humid and sub-humid parts of the landscape of Ethiopia where there are various research outputs highlighting this issue? How can these be considered in future similar research work as this one?

  We thank the referee for the comments. We believe that the uncertainty arising from lateral and subsurface flow is negligible in our results. This is because i) we applied the model only to the root zone (upper 60 cm) where unsaturated flow dominates the agro-hydrological fluxes, and ii) lateral flow is too slow to account for in a daily model at a coarse spatial resolution like 5 km x 5 km, thus change top soil layer water content from grid to grid. It would have an effect only close to the stream network where soil is more saturated and baseflow is produced, which again we do not resolve well with this course grid. Lateral soil water redistribution is important at spatial scales where topography provides sufficient gradients for subsurface flow, e.g., on the order of 1-100 m. Even at this finer scale where the surface slope is flat, in the top 60 cm soil root zone that we considered, the effect of lateral flow can still be low under a rainfed system. This is because the spatial distribution of rainfall and soil are nearly homogeneous, and thus the soil wets homogeneously unless there are conditions that lead to preferential flow (e.g., soil cracks and localized hardpans, as the referee mentioned). Lateral unsaturated flow in the root zone is more common and should be accounted for in simulations applied to irrigated plots, particularly under furrow and drip irrigation methods.

While authors have tried their best to validate the annual flow with literature data, the authors does not compare their result with other studies for yield.

We agree with the referee that we did not compare our yield results with those from other studies. This is simply because the attainable yield we computed is a relative value—the percentage of the potential yield (water-unlimited) that can be attained under the prevailing climate and soil conditions. We did not find any suitable similar study in the context of Ethiopia for comparison. However, we have tried to validate the attainable yield by correlating it to the actual total cereal production derived from the annual agricultural sample survey reports by the Central Statistical

Agency (CSA) of Ethiopia (Figure 4c). We believe that this provides a clue as to the reliability of the modelling framework we implemented.

The water limited attainable yield was estimated based on FAO equation. How certain is the result from the equation? Was it not possible to compare with experimental plots yield under various treatments? It is essential to support their findings with relevant literature and analysis of the result by comparing the increment or decrement of SMD or GWA with other studies too.

We thank the referee for the comments. The FAO productivity function is a basic and established relationship between climate and crop yield, explaining the relative yield as a function of the evapotranspiration ratio, the ratio of the actual to potential evapotranspiration. Crop coefficient-based models are built on this function and are used to assess crop yield responses to water-limited climate conditions (Foster and Brozović, 2018). For our stated purpose—assessing water-limited attainable yield to serve as a basis for reducing crop yield gaps in the rainfed farming systems in Ethiopia—we believe that the FAO water production function is a fair choice over more complex and data-intensive process-based crop models. We recognize the uncertainties arising from the simplicity of this model, but we believe that the level of uncertainty is acceptable, as we tried to demonstrate in our validations. We strongly suggest using process-based crop models if one aims to simulate absolute crop yields, especially at field and farm scales. We will stress these points in the revision of our manuscript. We agree with the referee that such analyses could benefit from validation with field experiments. However, the scale of this particular study makes it challenging to set up experimental plots that sufficiently cover the entire agricultural region. The time frame and resources allocated for this work did not allow for such efforts. Instead, we decided to rely on existing experimental data.

What was the problem collaborating with people and institutions in Ethiopia? There are institutions such as Ethiopia Agricultural Research Institute who do experiments under various agro-ecology for various crops. There was a possibility validating some of the results.

This study was a first level analysis of agroclimatic and hydrological effects on water-limited crop yield potential across Ethiopia. In a higher level analysis, one could focus on farm and landscape scales, at which point local experimental data from the Ethiopian Agricultural Research Institute and others would be interesting. We will pursue this in the future. However, it remains true that our methodology is not directly applicable to small scales which are represented by experimental crop yield data, as our approach is designed for large (national) scale assessment to estimate the 'relative yield'. For the purpose of validation of our results, we have used the best available crop data from the CSA, which we disaggregated from the zonal scale to the grid scale (Wakjira et al., 2021). For surface runoff, we collected published runoff plot measurement data at 17 locations, which we believe is fairly sufficient to test the performance of our model.

**Specific comments**

Page 4 line 110: why CHRIPS and ERA-5 were used? Why not other products?

In preliminary analyses we did explore a range of other rainfall products, and we concluded that CHIRPS is one of the most suitable daily rainfall data for Ethiopia and a large part of East Africa, offering one of the highest spatial resolutions, temporal continuity, and record lengths (e.g., Ahmed et al., 2024; Bayissa et al., 2017; Dinku et al., 2018; Gebrechorkos et al., 2018; Musie et al., 2019). For temperature and other climate variables, ERA5-Land provides better spatial resolution (9 km x 9 km) compared to other products. We performed bias-correction to ERA5-Land temperature and downscaled it to 5 km x 5 km grid resolution over Ethiopia (Wakjira et al., 2023).

Page 9 section 2.4 Assessment of Green water availability and its yield potential seems like result rather than methodology. So I recommend to put this section in the result part

Thank you. We will consider this in the revision.

Page 10, lines 27–29: The authors evaluated the simulations of AY [%] in terms of their correlation to variations in TCP [tonne y -1], however, the computed values of AY from the model are not mentioned in the manuscript. So, what are the order of magnitudes of these total crop production by showing the range under different agroclimate?

The total cereal production (TCP) is the sum of all cereal crops produced in Ethiopia. The 16-year (1995-2010) average TCP across the study region ranges from 28 tonnes to 24,000 tonnes.

Page 10 section 3.1: the wording of "observation" used in the section is misleading.

Thank you. We will revise our use of the terminology.

Page 12 Figure 4a and b. The performance evaluation of the model for runoff and actual evapotranspiration, the study used R2 and NSE for runoff and R2 for actual evapotranspiration. Why NSE is not used for actual evapotranspiration.

We thank the referee for raising this comment. The corresponding NSE for evapotranspiration is 0.8. We will add this to Figure 12b in the revision.

Figure 4: The 17 surface runoff data points obtained from the literature should be located on the map to see their spatial distributions.

Thank you. We will add the locations of the surface runoff plot data to Figure 12a in the revision.

Page 13 line 3-4: The reference climatology of growing season GWA and water-limited yield across the RFA region based on the computed SMD and AY values is presented, and considering alfalfa reference grass (Ky = 1.1), what is the source for this value?

We thank the referee for the comment. The yield response factor (Ky) values for alfalfa grass were taken from the FAO Irrigation and Drainage Paper 56 (Allen et al., 1998). We will indicate this in the revision.

Page 26 line 485: There are various research works that show the infiltration of the soil is very high compared to rainfall intensity. The problem is the hardpan formation limiting the infiltration through the root zone. The hardpan formation at lower depth facilitates the later subsurface flow with the terrain high slope. Authors need to mention this as part of their recommendation.

Thank you. We believe that this is true for specific locations that are subjected to the conditions that result in hardpan formation such as high heavy machinery traffic like in highly mechanized agricultural lands, flat areas with heavy clay soils exposed to inundation followed by drought conditions, etc. Also limiting infiltration are very steep terrain gradients, which as mentioned above we cannot capture with the coarse spatial resolution of our framework. We will mention these limitations in the revision.

**References**

Ahmed, J. S., Buizza, R., Dell'Acqua, M., Demissie, T. and Pè, M. E.: Evaluation of ERA5 and CHIRPS rainfall estimates against observations across Ethiopia, Meteorol. Atmos. Phys., 136(3), doi:10.1007/s00703-024-01008-0, 2024.

Allen, R. G., Pereira, L. S., Raes, D. and Smith, M.: Crop Evapotranspiration, Rome., 1998.

Arnold, J. G., Moriasi, D. N., Gassman, P. W., Abbaspour, K. C., White, M. J., Srinivasan, R., Santhi, C., Harmel, R. D., Van Griensven, A., Van Liew, M. W., Kannan, N. and Jha, M. K.: SWAT: Model use, calibration, and validation, Trans. ASABE, 55(4), 1491–1508, 2012.

Bayissa, Y., Tadesse, T., Demisse, G. and Shiferaw, A.: Evaluation of satellite-based rainfall estimates and application to monitor meteorological drought for the Upper Blue Nile Basin, Ethiopia, Remote Sens., 9(7), 1–17, doi:10.3390/rs9070669, 2017.

Čerkasova, N., White, M., Arnold, J., Bieger, K., Allen, P., Gao, J., Gambone, M., Meki, M., Kiniry, J. and Gassman, P. W.: Field scale SWAT+ modeling of corn and soybean yields for the contiguous United States: National Agroecosystem Model Development, Agric. Syst., 210(June), doi:10.1016/j.agsy.2023.103695, 2023.

Dinku, T., Funk, C., Peterson, P., Maidment, R., Tadesse, T., Gadain, H. and Ceccato, P.: Validation of the CHIRPS satellite rainfall estimates over eastern Africa, Q. J. R. Meteorol. Soc., 144, 292–312, doi:10.1002/qj.3244, 2018.

Foster, T. and Brozović, N.: Simulating Crop-Water Production Functions Using Crop Growth Models to Support Water Policy Assessments, Ecol. Econ., 152(March), 9–21, doi:10.1016/j.ecolecon.2018.05.019, 2018.

Gebrechorkos, S. H., Hülsmann, S. and Bernhofer, C.: Evaluation of multiple climate data sources for managing environmental resources in East Africa, Hydrol. Earth Syst. Sci, 22, 4547–4564, doi:10.5194/hess-22-4547-2018, 2018.

Hsiao, T. C., Heng, L., Steduto, P., Rojas-Lara, B., Raes, D. and Fereres, E.: Aquacrop-The FAO crop model to simulate yield response to water: III. Parameterization and testing for maize, Agron. J., 101(3), 448–459, doi:10.2134/agronj2008.0218s, 2009.

Jaafar, H. H., Ahmad, F. A. and El Beyrouthy, N.: GCN250, new global gridded curve numbers for hydrologic

modeling and design, Sci. Data, 6(1), 1–9, doi:10.1038/s41597-019-0155-x, 2019.

Jones, J. W., Hoogenboom, G., Porter, C. H., Boote, K. J., Batchelor, W. D., Hunt, L. A., Wilkens, P. W., Singh, U., Gijsman, A. J. and Ritchie, J. T.: The DSSAT cropping system model., 2003.

Musie, M., Sen, S. and Srivastava, P.: Comparison and evaluation of gridded precipitation datasets for streamflow simulation in data scarce watersheds of Ethiopia, J. Hydrol., 579(September), 124168, doi:10.1016/j.jhydrol.2019.124168, 2019.

Qi, J., Lee, S., Zhang, X., Yang, Q., McCarty, G. W. and Moglen, G. E.: Effects of surface runoff and infiltration partition methods on hydrological modeling: A comparison of four schemes in two watersheds in the Northeastern US, J. Hydrol., 581(November 2019), 124415, doi:10.1016/j.jhydrol.2019.124415, 2020.

Raes, D., Steduto, P., Hsiao, T. C. and Fereres, E.: Aquacrop-The FAO crop model to simulate yield response to water: II. main algorithms and software description, Agron. J., 101(3), 438–447, doi:10.2134/agronj2008.0140s, 2009.

Ross, C. W., Prihodko, L., Anchang, J., Kumar, S., Ji, W. and Hanan, N. P.: HYSOGs250m, global gridded hydrologic soil groups for curve-number-based runoff modeling, Sci. data, 5, 180091, doi:10.1038/sdata.2018.91, 2018.

Sishu, F. K., Tilahun, S. A., Schmitter, P. and Steenhuis, T. S.: Revisiting the Thornthwaite Mather procedure for baseflow and groundwater storage predictions in sloping and mountainous regions, J. Hydrol. X, 24(March), 100179, doi:10.1016/j.hydroa.2024.100179, 2024.

Steduto, P., Hsiao, T. C., Raes, D. and Fereres, E.: Aquacrop-the FAO crop model to simulate yield response to water: I. concepts and underlying principles, Agron. J., 101(3), 426–437, doi:10.2134/agronj2008.0139s, 2009.

Steenhuis, T. and Molen, W.: The TM procedure as a simple engineering method to predict recharge, J. Hydrol., 84, 221–229, 1986.

USACE: Hydrologic Modeling System Technical Reference Manual., 2000.

Wakjira, M. T., Peleg, N., Anghileri, D., Molnar, D., Alamirew, T., Six, J. and Molnar, P.: Rainfall seasonality and timing: implications for cereal crop production in Ethiopia, Agric. For. Meteorol., 310, 108633, doi:10.1016/J.AGRFORMET.2021.108633, 2021.

Wakjira, M. T., Peleg, N., Burlando, P. and Molnar, P.: Gridded daily 2-m air temperature dataset for Ethiopia derived by debiasing and downscaling ERA5-Land for the period 1981–2010, Data Br., 46, 108844, doi:10.1016/j.dib.2022.108844, 2023.

Westenbroek, M. S., Kelson, V. a., Dripps, W. R., Hunt, R. J. and Bradbury, K. R.: SWB — A Modified Thornthwaite-Mather Soil-Water- Balance Code for Estimating Groundwater Recharge, U.S. Geol. Surv. Tech. Methods 6-A31, 60, 2010.

Young, R. A., Onstad, C. A., Bosch, D. D. and Anderson, W. P.: AGNPS: A nonpoint-source pollution model for evaluating agricultural watersheds, J. Soil Water Conserv., 44(2), 168–173, 1989.

Zhao, C., Liu, B., Xiao, L., Hoogenboom, G., Boote, K. J., Kassie, B. T., Pavan, W., Shelia, V., Kim, K. S., Hernandez-Ochoa, I. M., Wallach, D., Porter, C. H., Stockle, C. O., Zhu, Y. and Asseng, S.: A SIMPLE crop model, Eur. J. Agron., 104(February), 97–106, doi:10.1016/j.eja.2019.01.009, 2019.

---

## Author Response (AR1)

Prof. Dr. Wouter Buytaert

Editor, Hydrology and Earth System Sciences (HESS)

14 September 2024

**Re: Submission of revised manuscript (HESS-2024-37)**

Dear Prof. Dr. Buytaert,

My co-authors and I thank you very much for handling the editorial process of our manuscript entitled '*Green water availability and water-limited crop yields under a changing climate in Ethiopia*'. We extend our gratitude to the anonymous reviewer and Rike Becker for their time and expertise to review our work, and for their constructive comments and suggestions. We are also grateful for the opportunity to revise and submit our manuscript.

As requested, we have made major revisions to the manuscript based on the comments and suggestions provided, as well as the concerns raised by both reviewers. The following major changes have been made to the manuscript:

- The choice of the SCS-CN method and water production function have been contextualized and clarified (*methods section*)
- The robustness of the modelling framework has been further evaluated for the agrohydrological variables simulated: surface runoff, soil moisture, actual evapotranspiration, and water-limited attainable yield (AY), using independent data from local studies and global database (*methods and result sections*).
- The findings for the changes in soil moisture deficit and AY have been discussed in views of related previous studies in a newly introduced section (*Discussion section*)
- The limitations of the modelling framework and possible uncertainties in the results have been elaborated. In particular, the SCS-CN method was compared to a water balance model (*Discussion section*)
- The figure styles have been revised, the use of figures in the main text have been revised, new figures have been produced and presented either in the main text or in the supplementary materials (*methods, result and discussion*)

The changes made have been indicated in the manuscript with tracked change.

Enclosed are our responses (*in blue*) to the reviewers' comments (*in black*), providing clarifications and how we addressed the comments in the revised manuscript. The *page and line numbers mentioned in the responses refer to the manuscript with tracked changes*.

Kind regards,

Mosisa Tujuba Wakjira,
On behalf of all co-authors

**1. Reviewer 1**

**General Comment**

The study focuses on the effects of climate change on green water availability and water-limited attainable yields (AY) for major cereal crops in Ethiopia. It develops an agro hydrological Modelling framework to simulate climatic-hydrological-crop interactions for the reference year (1981-2010) and future periods (2020-2099) under different greenhouse gas emission scenarios. The study discuss the importance of green water management practices to improve crop yields, especially in rain fed agricultural systems. It identifies a significant gap between actual and water-limited yields, emphasizing the need for integrated management strategies to overcome yield-reducing factors. The findings suggest that future changes in AY vary across regions, emission scenarios, and growing seasons, with temperature increase playing a key role in driving changes. The study concludes by recommending the adoption of climate-smart agricultural practices and the use of agro hydrological Modelling for informed decision-making in agricultural management planning. It is well written and have a significant contribution in green water management beyond the study area.

We thank the referee very much for the positive feedback.

The modelling approach used in this study provides valuable insights about the future without using process-based modelling. However, the approach is subjected to uncertainties and assumptions that may impact the accuracy of the results. Can the authors clarify as limitation or future research considerations in the following issues?

1.1. The partitioning of rainfall to runoff is based on the CN approach which is completely empirical and needs locally adjusted CN based on soil, land use and hydrologic conditions. How is this affecting the competition where there are no locally contextualized CN values?

We thank the reviewer for this comment and question. We have addressed this in the revised manuscript by analyzing the uncertainties associated with the limitations of the CN method, comparing it to a water balance (WB) approach based on maximum soil infiltration capacity (see pages 31-32, Lines 612-636 of the manuscript with tracked changes). The WB-based model we used assumes that runoff occurs only when the rainfall on a given day exceeds the soil's available storage capacity, defined as the difference between the water content at saturation and the soil water content from the previous day. We found negligible differences in soil moisture, actual evapotranspiration, and AY simulated using the CN-based and WB-based CHC models. However, the WB-based model does not effectively distinguish between surface runoff (Q) and deep percolation (Dp) as shown in Figure S16 in the revised supplementary material.

1.2. What about using other partitioning approaches such as Thornthwaite and Mather soil moisture water balance model widely applied in the area?

The WB-based model mentioned in number 1.1 is similar to the working principle of the Thornthwaite and Mather (TM) model (Sishu et al., 2024; Steenhuis and Molen, 1986) in terms of rainfall-runoff partitioning. Based on our evaluation (pages 31-32, Lines 612-636), we conclude that the CN-based model, despite its empirical nature, is a better choice for agrohydrological applications compared to the Thornthwaite and Mather model, which is a monthly soil moisture and groundwater recharge model. The TM model does not account for soil properties and land surface conditions that influence rainfall-runoff processes, as it relies solely on rainfall and potential evapotranspiration conditions (Sishu et al., 2024). Our agrohydrological modeling framework provides reliable estimates of key soil water balance components, including surface runoff, on a daily time step. These estimates are highly relevant for water management planning, such as soil erosion control, flood (spate) irrigation design, and agricultural drainage system design among others, where surface runoff estimation is essential. To keep the manuscript concise, we did not include this discussion in the manuscript, except for a brief reference to the TM model in a related section. We hope the reviewer understands our decision.

1.3. Can authors clarify how the lateral and subsurface flow likely affect the uncertainty? This kind of flow is very important in humid and sub-humid parts of the landscape of Ethiopia where there are various research outputs highlighting this issue? How can these be considered in future similar research work as this one?

We believe that the uncertainty arising from lateral and subsurface flow is negligible in our results, particularly in the upper 60 cm soil layer and at a coarse spatial resolution, such as 5 km x 5 km on a daily time scale. The effects of lateral and subsurface flow become relevant and need to be considered at finer spatial resolutions, such as on the order of tens of meters. These processes are better captured by infiltration-based soil water flux models, such as the Richards equation. We have briefly clarified this in the revised manuscript (page 32, Lines, 612-620; 630-632).

1.4. While authors have tried their best to validate the annual flow with literature data, the authors does not compare their result with other studies for yield. The water limited attainable yield was estimated based on FAO equation. How certain is the result from the equation? Was it not possible to compare with experimental plots yield under various treatments? It is essential to support their findings with relevant literature and analysis of the result by comparing the increment or decrement of SMD or GWA with other studies too.

Thank you. In the revised manuscript (see pages 12-13, Figure 3d, and Lines 291-304), we now have compared the simulated water-limited attainable yield (AY) with independent AY estimates at 45 locations (Table S4) across the RFA region. The independent AY estimates were derived from published field trials, calibrated crop model simulations, and the Global Yield Gap Atlas (www.yieldgap.org). Despite potential uncertainties in these independent estimates and the scale differences between the simulated (grid scale) and independent AY (point scale), we found a strong correlation, suggesting the robustness of the FAO water productivity function for our

application. We also supported our findings with other relevant literature (see pages 29-30, Section 4.2) on climate-driven changes in green water availability and crop yield.

1.5. What was the problem collaborating with people and institutions in Ethiopia? There are institutions such as Ethiopia Agricultural Research Institute who do experiments under various agro-ecology for various crops. There was a possibility validating some of the results.

This study represents a first-level analysis of agroclimatic and hydrological effects on water-limited crop yield potential across Ethiopia. In a more detailed, higher-level analysis, one could focus on farm and landscape scales, where local experimental data from the Ethiopian Agricultural Research Institute and other sources would be valuable. We are open to extending this work in collaboration with institutions in Ethiopia, considering more specific local conditions, for example, to identify yield gaps and explore how they can be bridged through effective green water management. For the validation of our results, we believe we have used the best available crop and agrohydrological data.

**Specific comments**

1.6. Page 4 line 110: why CHRIPS and ERA-5 were used? Why not other products?

In preliminary analyses, we explored a range of other rainfall products and concluded that CHIRPS is one of the most suitable daily rainfall datasets for Ethiopia and much of East Africa, offering one of the highest spatial resolutions, temporal continuity, and record lengths (e.g., Ahmed et al., 2024; Bayissa et al., 2017; Dinku et al., 2018; Gebrechorkos et al., 2018; Musie et al., 2019). For temperature and other climate variables, ERA5-Land provides better spatial resolution (9 km x 9 km) compared to other products. We performed bias-correction to ERA5-Land temperature and downscaled it to 5 km x 5 km grid resolution over Ethiopia (Wakjira et al., 2023). In the interest of keeping the manuscript concise, we did not include this discussion. We hope the reviewer agrees with this decision.

1.7. Page 9 section 2.4 Assessment of Green water availability and its yield potential seems like result rather than methodology. So I recommend to put this section in the result part

Thank you. We have moved the text to result section (page 19, Lines 389-392) and associated Figure 3 of the original manuscript to the supplementary materials as Figure S4.

1.8. Page 10, lines 27–29: The authors evaluated the simulations of AY [%] in terms of their correlation to variations in TCP [tonne y -1], however, the computed values of AY from the model are not mentioned in the manuscript. So, what are the order of magnitudes of these total crop production by showing the range under different agroclimate?

The 16-year (1995-2010) average TCP across the study region ranges from 28 tonnes to 24,000 tonnes. We have indicated this on page 16, Lines 342-343.

1.9. Page 10 section 3.1: the wording of "observation" used in the section is misleading.

Thank you. In the revised manuscript, we used the term 'global dataset' for soil moisture and actual evapotranspiration, instead of 'observation'.

1.10. Page 12 Figure 4a and b. The performance evaluation of the model for runoff and actual evapotranspiration, the study used R2 and NSE for runoff and R2 for actual evapotranspiration. Why NSE is not used for actual evapotranspiration.

In the revised version, we used Pearson's correlation coefficient and root mean squared deviation as comparison metrics instead of $R^2$ and NSE.

1.11. Figure 4: The 17 surface runoff data points obtained from the literature should be located on the map to see their spatial distributions.

Thank you. We have plotted the locations of runoff plots and experimental sites for yield data (page 13, Figure 3a and 3d).

1.12. Page 13 line 3-4: The reference climatology of growing season GWA and water-limited yield across the RFA region based on the computed SMD and AY values is presented, and considering alfalfa reference grass (Ky = 1.1), what is the source for this value?

The yield response factor (Ky) values for alfalfa reference grass was taken from the FAO Irrigation and Drainage Paper 56 (Allen et al., 1998). We have indicated this in the revised manuscript with tracked change (page 16, Line 350)

1.13. Page 26 line 485: There are various research works that show the infiltration of the soil is very high compared to rainfall intensity. The problem is the hardpan formation limiting the infiltration through the root zone. The hardpan formation at lower depth facilitates the later subsurface flow with the terrain high slope. Authors need to mention this as part of their recommendation.

Thank you. This comment is related to the comment on 1.3 above. We hope that have addressed it on page 32, Lines, 612-620; 630-632 of the revised manuscript.

**2. Reviewer 2:**

**General Comment**

The study describes the development and application of a simple Climate-Hydrological-Crop model to assess the impact of climate change on green water availability (GWA) and water-limited attainable yields (AY). The authors highlight the need for such an approach to simplify climate impact assessments for agricultural production, which oftentimes require complex and data-intensive process-based or statistical crop-climate models. After a detailed description of the methodology and the data used, the authors apply the framework to rainfed agricultural regions in Ethiopia and quantify GWA and AY under current and three potential future climate scenarios. Finally, the authors compare temperature vs. moisture related impacts to assess the main climatic cause of future yield changes, which could help agricultural practitioners to identify effective adaptation measures.

The paper is in most parts well written (see specific comments on manuscript structure further below). It starts with a comprehensive introduction which highlights the need for such a simple modelling approach. The methodology is explained thoroughly and gives all required information to reproduce the results. The data sets used to setup the model are all freely and globally available, which makes the presented approach widely applicable and adoptable by other researchers. The topic is relevant, particularly for regions with dominating rainfed agricultural production, where yields are highly sensitive to future climatic changes.

However, uncertainties of the framework should be discussed in more detail, and the results should be validated more thoroughly (see specific comments on uncertainty and validation below). In addition, I suggest restructuring parts of the manuscript and reconsidering/limiting the choice of figures to help the reader focus on the main objectives (see specific comments on figures below). I therefore recommend to address the following points before publication.

We thank Rike Becker very much for the constructive feedback.

**Uncertainties**: I understand that the main aim of this study is to develop a simple model, without very complex process representations. Yet, the uncertainties of that model should be discussed in more detail. I see potential uncertainties in the following points:

2.1. Equation 10: Is a single evaporative stress index (ESI) per season calculated? Crops react differently to stresses depending on their phenological phase (i.e. yield will be affected differently if stress hits early or late in growing season). Taking one single stress value for the entire season might introduce uncertainty.

Yes, ESI was calculated seasonally. Our modelling framework falls under what is categorized in crop and agroecosystem modelling as 'crop coefficient models' where the FAO water production function coupled with a

hydrological model to simulate crop yield response to seasonal climate (Foster and Brozović, 2018; He et al., 2022; Schwartz et al., 2020). We, however, agree with Rike that the sensitivity of yield to water stress depends on the phenological phase at which the stress occurs. We have discussed this and related issues as limitations in the discussion section of the revised manuscript (see pages 32-33, Lines 637-645 in the manuscript with tracked changes).

2.2. Is the assumption correct, that all four crops are grown equally in both seasons (same sowing and harvesting times) and that all crops are grown in each region/pixel? If not, the climatic stresses would have to be calculated based on the exact months they are grown and for the exact region. Averaging over a common period and a common region for all crops might introduce another point of uncertainty. How robust are the results for each specific crop?

We appreciate the points raised by Rike concerning the growing seasons and growing regions we considered. Considering that temperature variation between the two seasons (Meher and Belg) is small, the thermal condition does not limit production of any crop in either season. Moisture is the main limiting factor, especially during the Belg season. Our aim here was to examine the potential AY in both seasons to provide insight into the possible intensification of production, particularly during the Belg growing season.

As general as our assessment is, we chose to calculate the climatic stress for the main rainy months instead of the exact growing period (from sowing to harvesting date) because these vary significantly from year to year within the rainy seasons. The sowing and harvesting dates of each crop within both seasons can vary due to the interannual variability of the onset of the rainy season (Wakjira et al., 2021). The growing periods can also vary due to non-climatic drivers that affect farmers' schedules, for example, labor availability for pre-sowing operations like tillage, and the timing of agricultural input distribution, especially fertilizer, and seeds (FEWS NET, 2022; Spielman et al., 2011), which can delay sowing dates. We have highlighted this under the uncertainty and limitation section (Pages 32-33, Lines 645-649) in the revised manuscript

**Each crop grown in each pixel?** No, we mapped the potential AY at every pixel (Figure 5) for the reference grass (alfalfa), not for specific crops. This provides a general picture of the green water potential for crop production in both seasons. The changes in AY for the four crops were mapped only for agroecologically suitable pixels, as we analyzed in our other work (Wakjira, 2024).

2.3. Large uncertainties in the climate model predictions are averaged to a median value. Potential upper and lower bounds could be mentioned.

Thanks for this comment. We have mentioned this in the revised manuscript (Page 19, Lines 406-408).

2.4. Soil moisture is taken as a metric to estimate green water availability (= one of the main objectives of this study). How robust are the modelled soil moisture results? Fig. 3 shows a significant spread in the relationship of yield

to soil moisture deficits (~40% difference in yield estimates (~50%-90%) at SMD 40%). Can you elaborate on this uncertainty in the discussion section?

The spread in the AY versus SMD relationship is something we expected. It mainly reflects the spatial variability in soil properties and climate conditions. A low AY for 40% SMD can be associated with sandy soil in a dry climate, while a high AY for the same SMD can be linked to loam soil in a humid climate. This has been clarified in the revision (Page 19, Lines 389-392).

**Validation and robustness of results**: I agree with reviewer_#1 that the results should be validated more thoroughly.

2.5. Several studies on climate change impacts on crop yields in Ethiopia exists using more complex physically based (e.g. DSSAT or APSIM) or statistical crop-climate modelling approaches (e.g. Kassaye et al., 2021; Yang et al., 2023; many others are cited in the manuscript). Can these results be compared with the results of this study?

Thank you for this comment. A discussion sub-section has now been added (Pages 29-30).

2.6. Several remote sensing data sets for soil moisture exist. Can they be used to validate the simulated change in soil moisture and the simulated patterns? Like the assessment of ET, but also showing a spatial validation of the results.

The simulated soil moisture has now been evaluated by comparing it with four satellite-based products (Page 13, Figure 3b on page 13, also Figure S1 (for annual cycles) and Figure S3 (for spatial comparisons) in the supplementary materials).

2.7. A 5-year validation period is chosen. Are correlation coefficients of simulated and observed yields in Fig, 4c based on only 5 points (one point per year)? Can a longer time period be chosen? Did these years cover any extreme hot or dry years? How do the yield values respond to particularly hot/wet years (sensitivity of the model to extremes and climate variability)?

The correlation coefficients in Figure 4c (Figure 4 in the revised manuscript) were computed considering a 16-year (1995-2010) period. The TCP data is available for every cultivated pixel for the Meher season, which means, 16 data points per pixel (one point per year).

Both TCP and AY reflect the variability of climatic drivers -- rainfall and atmospheric evaporative demand as illustrated in Figuire R1 below. For example, the years 2002-2004 were drier than the average conditions during 1995-2010, with higher ETo in most parts of the RFA region. As expected, negative anomalies in both TCP and AY are evident during this period. In contrast, the years 2005-2008 were wetter than normal, ETo was lower, and both AY and TCP were stable. This highlights the robustness of the CHC modelling framework.

[Figure]

Figure R1: Comparisons of climatic variability, simulated water-limited attainable yield (AY), and observed total cereal production (TCP) from 1995 to 2010 at 45 locations (refer to Table) across the rainfed agricultural region of Ethiopia. The solid lines represent the average values across the locations for each year, while the shaded areas represent the lower and upper bounds (5th to 95th percentile) of the anomalies at those locations.

2.8. 4b - Validation of ET result: Regions were chosen randomly as stated in the text. Why? Could you align them with the climatic regions (sARD, dSHM, ...)? Region "South" includes extreme dry and extreme wet areas. A clearer assessment of potential under- or overestimation of ET for the regions listed in the results would be possible if regions would be the same.

We understand Rike's concern and appreciate the comments and suggestions. The simulated ETa and soil moisture are now evaluated for each climatic regime (see Figure 3b and c, page 13).

2.9. **Figures**: I suggest limiting the number of figures and focussing on the most relevant ones. Maybe these can be chosen according to the main research questions (lines 80-85), so that the answers to the research questions can be found in the figures? In my opinion, figures 3,6,7,8 could be moved to the supplementary material. I further recommend being more consistent with the figure display and carefully describe the figure in the text.

We appreciate these suggestions. In the revised manuscript, we have moved figures 3 and 8 to the supplementary materials (Figures S4 and S5). However, we decided to retain Figures 6 and 7 in the main manuscript. Figure 6 serves as the basis for the climate change impact analysis and related discussions, while Figure 7 presents the projected changes in soil moisture deficit during the main growing season -- one of the key research questions. We hope Rike agrees with our decision.

2.10. Sometimes only the plot for Meher is given in the manuscript (Fig. 10), other times both seasons are displayed (Fig. 5). Sometimes results for all crops are given (Fig. 10), other times only Maize and Teff results are shown (Fig. 9). Sometimes maps are shown for the entire agricultural region, other times regions are divided in administrative zones (Fig. 11). I would suggest aligning the style of the different plots and to make sure all figures for one specific point in the analysis (e.g. projected changes in AY for Meher and Belg) are plotted in the manuscript and not some in the manuscript, and others in the supplement.

We thank Rike for these comments. We have revised the figures to ensure consistence of the styles for those produced for similar purpose (Figures S6-S13).

Among the several figures produced in our analyses, we selected the most important ones for presentation in the manuscript and provided the others in the supplementary materials. For example, we focus more on Meher than Belg because Meher is the more important season, accounting for about 90% of the annual grain production. When we had to choose among the crops for our presentation, we focused on the top two crops produced in the country, which are teff and maize. For the climate scenario, we primarily show the intermediate scenario, and for the future period, if we had to be selective, we presented the mid-century period. We presented our results at the grid scale, except for the climate sensitivity analysis, which we chose to present at the administrative zone scale, as this is important information for practitioners to guide adaptation planning at the institutional level.

2.11. Manuscript structure: To help the reader focus on the main results of the study, I would suggest restructuring of the manuscript. Shortening and focussing of all section on the most important findings. Moving first two paragraphs of the discussion to the result section and the remaining part to the conclusion section. Using the discussion section to discuss the results of the previous analysis, highlighting interesting findings, discussing the strengths and weaknesses of the methodology, and discussing potential uncertainties of the results.

Thank you very much for the suggestions. We have made a few rearrangements and shortened some sections. However, we retained most of the discussion paragraphs in the same section to keep the conclusion concise. As mentioned earlier, we added a results discussion (Section 4.1), where we supported our findings with other related works, as well as a section (Section 4.2) highlighting the main uncertainties and limitations. We acknowledge and agree with Rike that the manuscript is lengthy, but we aimed to provide comprehensive information with practical implications for Ethiopian agriculture. This work is part of a PhD research project in which the first author assessed various aspects of climate, water, and agriculture across the RFA region of Ethiopia. We hope Rike understands our approach.

**References**

Ahmed, J. S., Buizza, R., Dell'Acqua, M., Demissie, T., and Pè, M. E.: Evaluation of ERA5 and CHIRPS rainfall estimates against observations across Ethiopia, Meteorol. Atmos. Phys., 136, https://doi.org/10.1007/s00703-024-01008-0, 2024.

Allen, R. G., Pereira, L. S., Raes, D., and Smith, M.: Crop Evapotranspiration, Irrigation and Drainage Paper No. 56, Rome, 326 pp., 1998.

Bayissa, Y., Tadesse, T., Demisse, G., and Shiferaw, A.: Evaluation of satellite-based rainfall estimates and application to monitor meteorological drought for the Upper Blue Nile Basin, Ethiopia, Remote Sens., 9, 1–17, https://doi.org/10.3390/rs9070669, 2017.

Dinku, T., Funk, C., Peterson, P., Maidment, R., Tadesse, T., Gadain, H., and Ceccato, P.: Validation of the CHIRPS satellite rainfall estimates over eastern Africa, Q. J. R. Meteorol. Soc., 144, 292–312, https://doi.org/10.1002/qj.3244, 2018.

Foster, T. and Brozović, N.: Simulating Crop-Water Production Functions Using Crop Growth Models to Support Water Policy Assessments, Ecol. Econ., 152, 9–21, https://doi.org/10.1016/j.ecolecon.2018.05.019, 2018.

Gebrechorkos, S. H., Hülsmann, S., and Bernhofer, C.: Evaluation of multiple climate data sources for managing environmental resources in East Africa, Hydrol. Earth Syst. Sci, 22, 4547–4564, https://doi.org/10.5194/hess-22-4547-2018, 2018.

He, J., Ma, B., and Tian, J.: Water production function and optimal irrigation schedule for rice (Oryza sativa L.) cultivation with drip irrigation under plastic film-mulched, Sci. Rep., 12, 1–12, https://doi.org/10.1038/s41598-022-20652-3, 2022.

Musie, M., Sen, S., and Srivastava, P.: Comparison and evaluation of gridded precipitation datasets for streamflow simulation in data scarce watersheds of Ethiopia, J. Hydrol., 579, 124168, https://doi.org/10.1016/j.jhydrol.2019.124168, 2019.

Schwartz, R. C., Domínguez, A., Pardo, J. J., Colaizzi, P. D., Baumhardt, R. L., and Bell, J. M.: A crop coefficient – based water use model with non-uniform root distribution, Agric. Water Manag., 228, 105892, https://doi.org/10.1016/j.agwat.2019.105892, 2020.

Sishu, F. K., Tilahun, S. A., Schmitter, P., and Steenhuis, T. S.: Revisiting the Thornthwaite Mather procedure for baseflow and groundwater storage predictions in sloping and mountainous regions, J. Hydrol. X, 24, 100179, https://doi.org/10.1016/j.hydroa.2024.100179, 2024.

Steenhuis, T. and Molen, W.: The TM procedure as a simple engineering method to predict recharge, J. Hydrol., 84, 221–229, 1986.

Wakjira, M. T.: Understanding the impacts of climate variability and change on rainfed crop production in Ethiopia, ETH Zurich, 244 pp., https://doi.org/10.3929/ethz-b-000672129, 2024.

Wakjira, M. T., Peleg, N., Burlando, P., and Molnar, P.: Gridded daily 2-m air temperature dataset for Ethiopia derived by debiasing and downscaling ERA5-Land for the period 1981–2010, Data Br., 46, 108844, https://doi.org/10.1016/j.dib.2022.108844, 2023.

---

## Author Response (AR2)

Prof. Dr. Wouter Buytaert

Editor, Hydrology and Earth System Sciences (HESS)

02 December 2024

**Re: Submission of revised manuscript (HESS-2024-37)**

Dear Prof. Dr. Buytaert,

On behalf of all the co-authors, I would like to thank you for the opportunity for revise and submit our manuscript entitled '*Green water availability and water-limited crop yields under a changing climate in Ethiopia*' (HESS-2024-37). I also forward our sincere gratitude to Seifu Tilahun (reviewer 1) and Rike Becker (reviewer 2) for their unlimited commitment to devote their time to review our work, and for their constructive comments and suggestions that improved our manuscript.

In this round of revision, we have made the following minor revisions to the manuscript based on the suggestions we recieved from our reviewer Rike Becker:

- A brief clarification on the figures we prioritized to present in the manuscript has been added
- The consistency of nomenclatures used in the manuscript has been checked and updated throughout the paper
- The legend of Figure 10 has been updated to indicate the reference of the changes in AY
- The unpublished work has been replaced by its published version in the revised manuscript

The changes made have been indicated in the manuscript with tracked change.

Enclosed are our responses (*in blue*) to the reviewer's comments (*in black*). The page and line numbers mentioned in the responses refer to the manuscript with tracked changes.

Kind regards,

Mosisa Tujuba Wakjira,
On behalf of all co-authors

**General Comment**

The authors have thoroughly revised their manuscript. A newly added chapter on potential uncertainties and limitations of the studies now substantially extend the discussion section and discusses shortcomings of the analysis. Figures were updated and/or moved for conciseness. The results are now compared to other similar studies in Ethiopia and agreements or disagreements are outlined in more detail. I accept the authors' responses to my previous concerns and only have a few minor comments:

We thank very much Rike Becker for the thorough evaluation, constructive feedback and invaluable suggestions.

Point 2.10: I understand the authors' reasoning behind the selection of the figures (focussing on Meher instead of Belg). A very short explanation for the reader, why you select these figures could help the reader understand your choice. Similar to your answer to my question.

Thank you very much for this suggestion. We have briefly added this to the first paragraph of section 3.3.2 (Lines 398-399, page 18). The paragraph now reads (In italics are the added sentences): "In the face of the expected warmer and mostly wetter climate across the RFA region of Ethiopia, the GWA is likely to increase. *Here, we present the expected changes in GWA in Meher (Figure 8), as an estimated 88 % of the total grain production in Ethiopia occurs during this season. The expected changes in Belg are shown in Figure S5 of the supplementary materials*. In Meher, SMD is expected to decrease by up to 5 % over a large part of the region in the 2030s under all emission scenarios (Figure 8). However, in the 2060s and 2080s, SMD is projected to increase by 3-15 % over the western, southwestern, and southern parts under the low and intermediate emission scenarios. This is expected, given the projected minimal change in rainfall under the warming climate over these regions".

Double check the consistency of nomenclature: sometimes capitals are used, sometimes small letters. E.g. 'belg' vs. 'Belg'. 'meher' vs. 'Meher'.

Thank you. We have checked this and made corrections throughout the manuscript.

Fig. 10: what do you mean by 'data1' in legend of Fig. 10? Replace with 'Reference AY' or something similar.

Thank you for pointing this out. We have replaced 'data1' with 'ref' in Fig. 10.

Line 449: check the journal policy if you can cite your own unpublished work. I am not sure, if this is allowed. Some journals have a particular policy on this.

Thanks. The paper that was under review has now been published. We have used the published version in the revised manuscript.